# Unbiased Gradient Estimation for Event Binning via Functional Backpropagation

**Jinze Chen, Wei Zhai\*, Han Han, Tiankai Ma, Yang Cao, Bin Li, Zheng-Jun Zha**
MoE Key Laboratory of Brain-inspired Intelligent Perception and Cognition,
University of Science and Technology of China
`{chjz@mail.,wzhai056@,hanh@mail.,tiankaima@mail.}ustc.edu.cn`
`{forrest,binli,zhazj}@ustc.edu.cn`

## Abstract

Event-based vision encodes dynamic scenes as asynchronous spatio-temporal spikes called events. To leverage conventional image processing pipelines, events are typically binned into frames. However, binning functions are discontinuous, which truncates gradients at the frame level and forces most event-based algorithms to rely solely on frame-based features. Attempts to directly learn from raw events avoid this restriction but instead suffer from biased gradient estimation due to the discontinuities of the binning operation, ultimately limiting their learning efficiency. To address this challenge, we propose a novel framework for unbiased gradient estimation of arbitrary binning functions by synthesizing weak derivatives during backpropagation while keeping the forward output unchanged. The key idea is to exploit integration by parts: lifting the target functions to functionals yields an integral form of the derivative of the binning function during backpropagation, where the cotangent function naturally arises. By reconstructing this cotangent function from the sampled cotangent vector, we compute weak derivatives that provably match long-range finite differences of both smooth and non-smooth targets. Experimentally, our method improves simple optimization-based egomotion estimation with 3.2% lower RMS error and $1.57\times$ faster convergence. On complex downstream tasks, we achieve 9.4% lower EPE in self-supervised optical flow, and 5.1% lower RMS error in SLAM, demonstrating broad benefits for event-based visual perception. Source code can be found at `https://github.com/chjz1024/EventFBP`.

## 1 Introduction

Event-based visual perception has recently emerged as a novel paradigm to encode highly dynamic scenes with spatio-temporal event spikes. This paradigm shift brings new opportunities to high-speed and low-latency visual information processing, such as fine-grained optical flow estimation (Zhu et al., 2018; Gehrig et al., 2021b; Shiba et al., 2023; Hamann et al., 2024; Luo et al., 2024; Wan et al., 2024; 2025; Han et al., 2025), high-speed robot localization (Vidal et al., 2018; Zhou et al., 2021; Hines et al., 2025), and blurless video generation (Pan et al., 2019; Rebecq et al., 2019; Tulyakov et al., 2021; Wu et al., 2024; Xu et al., 2025; Liao et al., 2025).

To leverage conventional image processing pipelines, a common practice is to apply data binning techniques to convert irregular events into dense frames (Gallego et al., 2020), as shown in Figure 1. However, binning functions are inherently discontinuous, which truncates gradients at the frame level and forces most event-based algorithms to rely solely on frame-based features. Existing approaches often resort to smooth binning to restore differentiability to raw events, but such surrogates inevitably introduce bias in the gradients, leading to suboptimal learning efficiency.

This challenge reflects a broader issue in learning with discontinuous nonlinearities in neuromorphic computing, where spiking neuron models introduce non-differentiable functions (Eshraghian et al., 2023). Existing solutions, such as surrogate gradients (Neftci et al., 2019) or straight-through estimators (Yin et al., 2019), provide heuristic gradients, but they lack unbiasedness guarantees.

---

\*Correspondence to: Wei Zhai <`wzhai056@ustc.edu.cn`>

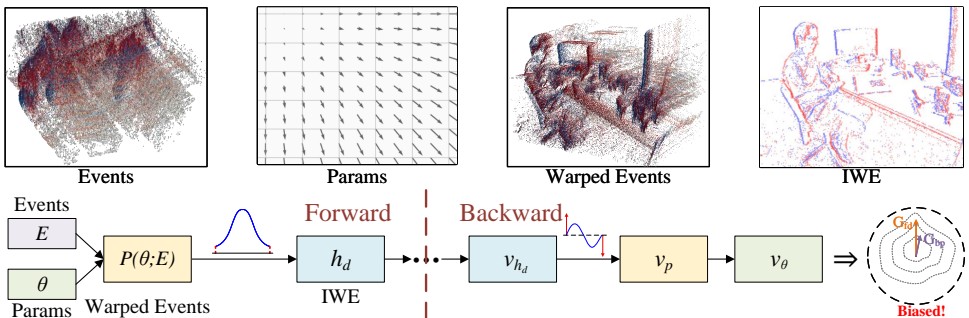

Figure 1: **Event binning and the gradient bias problem. Top:** Spatio-temporal event clouds (Events) are geometrically warped using motion parameters (Params) and aggregated into an Image of Warped Events (IWE) using a binning function. **Bottom Left (Forward):** The warping function $P(\theta; E)$ transforms input events $E$ and parameters $\theta$ into warped coordinates. These are processed by a discontinuous binning function $h_d$ to produce the IWE. **Bottom Right (Backward):** The adjoint (cotangent vector) of the IWE, denoted as $v_{h_d}$, is propagated back to update parameters. However, the discontinuity of $h_d$ results in non-computable Dirac delta functions when computing the gradient $v_p$ for the warped events. **Result:** The computed backpropagation gradient $G_{bp}$ deviates from the true finite difference gradient $G_{fd}$, shown in the contour plot as a "Biased!" estimation.

From a mathematical perspective, weak derivatives (Kuttler, 2017) generalize classical derivatives to discontinuous functions. While their pointwise values may be non-computable (*e.g.,* Dirac delta), their integrals are well-defined and can be computed via integration by parts. This observation is crucial: if we can computationally match the integral of weak derivatives, then the resulting gradient estimation recovers unbiasedness.

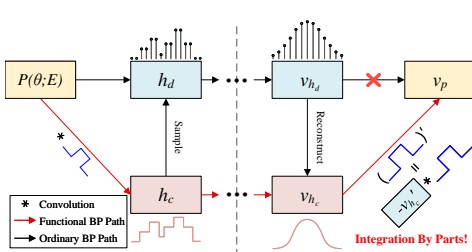

Figure 2: **The proposed Functional Backpropagation (FBP) framework.** To resolve discontinuities in the ordinary path (top, discrete binning $h_d$), we lift the operation to a functional space (bottom, continuous binning $h_c$). FBP bridges the two by reconstructing the continuous cotangent function $v_{h_c}$ from the discrete samples $v_{h_d}$. Using integration by parts, we replace the undefined Dirac delta evaluation with a convolution ($*$) of $v_{h_c}$ and the kernel derivative, synthesizing an unbiased gradient $v_p$.

To find such an integral, we dive into the backpropagation process and find that if we lift the binning function to the space of functionals, an integral of the respective gradient and the cotangent function naturally arises, which is provably sampled by the cotangent vector, as shown in Figure 2. Because cotangent functions naturally encode continuous motion flow, smooth priors can be utilized to reconstruct them from the cotangent vector irrespective of the forward pass. This allows us to derive an exact formula for the synthesized weak derivative of the binning function, which is provably shown to approximate long-range finite differences of arbitrary target functions. It also applies to continuous soft binning functions, where the synthesized gradients may help to skip local extremes.

To validate all the above claims, we first conduct analytical studies on simple tasks of event-based motion estimation, with an impressive result on the convergence speed and increased accuracy. We then conduct experiments on real-world tasks of optical-flow estimation and SLAM to show its wide applicability.

To summarize, our main contributions are:

1. We identify and formalize the fundamental issue of biased gradient estimation in event-based pipelines, stemming from the discontinuity of binning functions.

2. We propose a novel functional backpropagation framework that lifts binning functions into functional space, enabling gradient computation via weak derivatives. By applying integration by parts, our method avoids Dirac deltas and restores unbiasedness.

3. We derive an exact formula for the synthesized weak derivative, proving its equivalence to long-range finite differences for arbitrary functions, and show that it naturally generalizes to both discontinuous and smooth kernels.

4. We demonstrate consistent improvements across tasks: 3.7% lower RMS velocity error with $1.57\times$ faster convergence in controlled settings, 9.4% lower EPE in optical flow, and 5.1% lower RMS trajectory error in SLAM.

## 2 RELATED WORK

**Binning in Event-Based Vision.** In event-based vision, binning extends beyond its canonical form to support learning diverse visual representations. Two main types can be distinguished depending on where the information resides. The first focuses on binning weights, such as constants or event polarities (Maqueda et al., 2018; Luo et al., 2025), latest event timestamps (Lagorce et al., 2016; Ghosh & Gallego, 2025), or relative event intervals (Pan et al., 2019; Teng et al., 2022), which are primarily used to model irradiance change, motion, and gradients, respectively. These methods typically ignore binning gradients during learning and thus suffer from information loss. The second type focuses on binning locations, where events undergo parametric transformations, and parameters are recovered by optimizing sharpness metrics of the resulting event frames (Gallego et al., 2018; Gu et al., 2021; Shiba et al., 2024). While this approach captures complex spatio-temporal dynamics, it is fundamentally limited by the non-differentiability of the binning function. Our work is mainly concerned with this latter category.

**Learning with Discontinuous Nonlinearities.** There are three common approaches to restoring differentiability for discontinuous functions. The most direct is function relaxation (Huh & Sejnowski, 2018), but it alters the output and may compromise its physical meaning. Straight-through estimators (STE) (Yin et al., 2019) heuristically replace the gradient with a surrogate, while surrogate gradients (SG) (Neftci et al., 2019) introduce continuous relaxations of the true gradients. However, none of these methods guarantee unbiasedness, and their effectiveness often depends on domain expertise in choosing appropriate relaxations. Our method instead reconstructs a continuous relaxation of the cotangent function, providing a complementary alternative to surrogate gradients with the key property of unbiasedness.

**Weak and Functional Derivatives.** Our method draws on functional analysis, in particular weak derivatives and functional derivatives. Weak derivatives are defined through smooth test functions, where integration by parts provides an alternative characterization (Kuttler, 2017). Functional derivatives extend the notion of gradients to functionals, playing a central role in the calculus of variations (Frigyik et al., 2008). While these concepts are typically of theoretical interest due to computational intractability, we show that they naturally arise in backpropagation when lifting binning functions to the space of functionals, thereby enabling their practical use in learning.

## 3 METHODOLOGY

This section is organized in four parts: First, we extend ordinary backpropagation to functionals where an integral form naturally arises alongside the cotangent function, by deriving the chain rule for arbitrary functionals besides integrals encountered in the calculus of variations. Then we prove that an ordinary backpropagation pass samples the cotangent function with cotangent vectors, so that signal reconstruction techniques can be applied to compute the weak derivative of binning functions. Subsequently, we present a pseudocode implementation to bridge the gap between the theoretical derivation and practical application in forward and backward modes. Finally, we experimentally validate that the method approximates long-range finite differences for various loss functions, and the formal proof of unbiasedness is included in the appendix.

We start with a notational convention to distinguish between functionals and pointwise evaluation:

- Square brackets are used for functionals acting on elements of a function space; *e.g.,* $f[u]$.

- Parentheses are used for pointwise evaluation of ordinary functions; *e.g., , $g(x)$.*

### 3.1 FUNCTIONAL BACKPROPAGATION

A *functional* is defined by a rule, which associates a number with a function. Let $\mathcal{H}(X) := (X \to \mathbb{R})$ represent the space of real-valued functions defined on $X$, then a *functional* $f$ is defined as a mapping $u \in \mathcal{H}(X) \mapsto f[u]$, where $f[u]$ can itself be another function on space $\mathcal{H}(Y)$. In this definition, functionals can be chained and used to derive the chain rule.

Since functionals are functions that consume an entire function to give another function, their differentials are also functions. From the knowledge of the Fréchet derivative and the Riesz representation theorem in functional analysis, we have the following result:

**Definition 1.** *Assuming $f$ is a differentiable functional on vectors spaces from $\mathcal{H}(X)$ to $\mathcal{H}(Y)$, then the derivative of $f$ at $u \in \mathcal{H}(X)$ is represented by the derivative kernel function $\frac{\delta f[u](y)}{\delta u(x)}$, which satisfies the limit:*

$$\lim_{\|\delta u\| \to 0} \frac{\|f[u + \delta u] - f[u] - \int_X \frac{\delta f[u]}{\delta u(x)} \delta u(x) dx\|_{\mathcal{H}(Y)}}{\|\delta u\|_{\mathcal{H}(X)}} = 0. \tag{1}$$

With the above definition, the chain rule for functionals is described in terms of integrals.

**Theorem 1.** *Let $\mathcal{H}(X) \xrightarrow{f} \mathcal{H}(Y) \xrightarrow{g} \mathcal{H}(Z)$ where $f$ and $g$ are differentiable functionals, then the composite functional $g \circ f$ is also differentiable and has the representation:*

$$\frac{\delta g[f[u]](z)}{\delta u(x)} = \int_Y \frac{\delta g[f[u]](z)}{\delta f[u](y)} \frac{\delta f[u](y)}{\delta u(x)} dy. \tag{2}$$

When $X, Y, Z$ are finite sets, Theorem 1 becomes the familiar chain rule $J_{g \circ f} = J_g \cdot J_f$ where $J$ is the finite-dimensional Jacobian matrix, since integration becomes summation on finite sets. So any composable differentiable mappings have a chain rule, where every ordinary function induces a Jacobian-vector product and every functional induces an integral transform with the derivative kernel function.

To compute the Jacobians, classical backpropagation recursively computes $vJ_{g \circ f}$ using the associative property of matrix multiplication on Jacobians as $(vJ_g) \cdot J_f$, where $v \in \mathcal{H}(Z)$ is called a cotangent vector (Shi et al., 2024). The extension for infinite-dimensional functionals is a function $v(\cdot) \in \mathcal{H}(Z)$, and we call it the cotangent function. In analogy, we call the process of recursively computing the cotangent function using the chain rule **Functional Backpropagation (FBP)**.

### 3.2 SYNTHESIZING WEAK DERIVATIVES OF THE BINNING FUNCTION

An event-based algorithm accepts a group of events as input, defined as

$$\mathcal{E} = \{e_i = (t_i, x_i, y_i, p_i)\}_{i=1}^{N_e}, \tag{3}$$

where $e_i$ represents an event with timestamp $t_i$, pixel location $(x_i, y_i)$, and polarity $p_i \in \{-1, +1\}$ representing sign of brightness change, and $N_e$ is the number of events. A parametric transform maps $\mathcal{E}$ to D-dimensional weighted points

$$\mathcal{P} = \{e_i' = (x_{i_1}', \cdots, x_{i_D}', w_i')\}_{i=1}^{N_e}, \tag{4}$$

where $x_{i_d}'$ is the transformed location and $w_i'$ is the weight. The binning function $\boldsymbol{h}$ is a map from $\mathcal{P}$ to $\mathbb{R}^{W_1 \times \cdots \times W_D}$ where $W_d$ is the number of bins. The value at index $(j_1, \cdots, j_D)$ is defined as

$$h_{j_1, \ldots, j_D} = \sum_{i=1}^{N_e} w_i' \prod_{d=1}^D k_d \left( \frac{x_{i_d}' - j_d \Delta_d}{\Delta_d} \right), \tag{5}$$

where $k_d(\cdot)$ is the binning kernel with finite support and $\Delta_d$ is the binning width. This section provides an intuitive derivation of the formula for synthesizing the weak derivative $\frac{\partial h_{j_1, \cdots, j_D}}{\partial x_{i_d}'}$. To simplify the derivation, our discussion is restricted to the $D = 1$ case, but the result can be easily extended to arbitrary dimensions. A structured Theorem-Proof is provided in the appendix.

We restate the definition of 1D event binning function $\boldsymbol{h}$ with input points $\mathcal{P}$ as:

$$\boldsymbol{h}(\mathcal{P}) = (h_1(\mathcal{P}), \cdots, h_W(\mathcal{P})), \quad h_j(\mathcal{P}) = \sum_{i=1}^{N_e} w_i' k(\frac{x_i' - j\Delta}{\Delta}). \quad (6)$$

To learn the parameters, a scalar loss is constructed from the obtained $\boldsymbol{h}(\mathcal{P})$ using $f_d : \boldsymbol{h}(\mathcal{P}) \mapsto f_d[\boldsymbol{h}[\mathcal{P}]] \in \mathbb{R}$. A continuous binning function $h(\mathcal{P}) \in \mathcal{H}(\mathbb{R})$ can be obtained by replacing $j\Delta$ with a continuous parameter $x$:

$$h[\mathcal{P}](x) = \sum_{i=1}^{N_e} w_i' k(\frac{x_i' - x}{\Delta}), \quad (7)$$

A natural mapping exists from $h[\mathcal{P}]$ to $\boldsymbol{h}[\mathcal{P}]$ using the sampling functional $\mathcal{S} : h[\mathcal{P}] \mapsto \mathcal{S}[h[\mathcal{P}]] := (h(\Delta), \cdots, h(W\Delta)) = \boldsymbol{h}[\mathcal{P}]$. The sampling operator induces a functional $f_c := f_d \circ \mathcal{S}$.

We denote the space of discrete binning values as $\mathcal{H}_d$ and the space of continuous binning values as $\mathcal{H}_c$, and represent their relationship using the following commutative diagram:

$$\begin{array}{ccc} & \mathcal{H}_c & \\ {\scriptstyle h} \nearrow & \downarrow {\scriptstyle \mathcal{S}} & \searrow {\scriptstyle f_c} \\ \mathcal{P} \xrightarrow{\ \boldsymbol{h}\ } & \mathcal{H}_d \xrightarrow{\ f_d\ } & \mathbb{R} \end{array} \quad (8)$$

To compute $\frac{\partial f_c}{\partial x_i'}$, functional backpropagation computes the cotangent vector $v_{x_i'}$ from $v_f$ by:

$$v_h(x) = v_f \frac{\delta f_c}{\delta h(x)}, \quad v_{x_i'} = \int_{\mathbb{R}} v_h(x) \frac{\partial h(x)}{\partial x_i'} dx, \quad (9)$$

The cotangent function $v_h(\cdot)$ can also be computed during the normal backpropagation as $f_c = f_d \circ \mathcal{S}$. Using the chain rule, we have:

$$v_h(x) = \sum_{j=1}^{W} v_{h_j} \frac{\delta h_j}{\delta h(x)} = \sum_{j=1}^{W} v_{h_j} \delta(x - j\Delta), \quad (10)$$

where $v_{h_j}$ is the classical cotangent vector of $f_d$ during backward pass and $\delta(\cdot)$ is the Dirac delta. In signal processing terms, Equation (10) states that a normal backward pass of the binning function $\boldsymbol{h}$ surrogates the cotangent function with a Dirac comb modulated by the cotangent vector. Fixing $W\Delta$, this surrogate becomes exact as $\Delta \to 0$, and at this time, we have:

$$\lim_{\Delta \to 0} \frac{1}{\Delta} v_{h_j} = v_h(j\Delta). \quad (11)$$

This result connects the cotangent function and cotangent vector by the sampling rule, so we can use normal signal reconstruction methods to reconstruct the cotangent function and derive an exact formula for $\frac{\partial f_c}{\partial x_i'}$. Since cotangent functions naturally encode smooth motion flow, one simple solution is to replace $\delta(\cdot)$ in Equation (10) with another kernel $\frac{1}{\Delta} l(\frac{\cdot}{\Delta})$, then we have:

$$\tilde{v}_{x_i'} = \sum_{j=1}^{W} \int_{\mathbb{R}} v_{h_j} \frac{w_i'}{\Delta^2} l'(\frac{x - j\Delta}{\Delta}) k(\frac{x_i' - x}{\Delta}) dx \overset{(a)}{=} \sum_{j=1}^{W} w_i' \frac{\partial}{\partial x_i'} \kappa(\frac{x_i' - j\Delta}{\Delta}) v_{h_j}, \quad (12)$$

where $\tilde{v}_{x_i'}$ is the synthesized cotangent vector, (a) relies on integration-by-parts, $l'(x) = \frac{dl(x)}{dx}$ and

$$\kappa(x) = \int_{\mathbb{R}} l(y) k(x - y) dy = (l * k)(x). \quad (13)$$

In other words, the synthesized weak derivative of $h_j$ is

$$\frac{\partial \tilde{h}_j}{\partial x_i'} = w_i' \frac{\partial}{\partial x_i'} \kappa(\frac{x_i' - j\Delta}{\Delta}). \quad (14)$$

Comparing Equation (14) with the formal derivative $\frac{\partial h_j}{\partial x_i'} = w_i' \frac{\partial}{\partial x_i'} k(\frac{x_i' - j\Delta}{\Delta})$, we see that it's equal to a surrogate gradient of kernel $\kappa(\cdot)$. It's possible to demonstrate the unbiasedness of our method using the concept of weak derivatives, compared to other heuristic surrogates. In fact, it has finite-order approximation precision to long-range finite differences of arbitrary targets. Detailed proofs are provided in the appendix. For the multidimensional case in Equation (5), the result also holds when to replace $k_d$ with $\kappa_d$ for each dimension. Implementation is provided in the following section.

## 3.3 ALGORITHM IMPLEMENTATION

To simplify the implementation of FBP despite the theoretical density, we observe that our method only modifies the backward pass gradient accumulation. To demonstrate how different dimensions interact, a 2D example Algorithm 1 illustrates that the discontinuous kernel derivative is substituted with the synthesized weak derivative $\kappa'(\cdot)$, while the forward pass remains unchanged.

---

**Input:** Locations $\mathbf{x}'_i = (x'_i, y'_i)$, weights $w'_i$, Grid params ($W \times H$, spacing $\Delta$).
**Input:** Kernels: Binning $k$, Reconstruction $l$. **Optional:** Tangents $\dot{\mathbf{x}}'_i = (\dot{x}'_i, \dot{y}'_i)$, Adjoints $\bar{H}$.
**Output:** Frame $H \in \mathbb{R}^{W \times H}$. **Optional:** Tangent Frame $\dot{H}$, Gradients $\nabla_{\mathbf{x}'}\mathcal{L}$.

$\kappa(u) \leftarrow (l * k)(u); \quad \kappa'(u) \leftarrow \frac{d}{du}\kappa(u);$      // Synthesize derivative kernel

$H \leftarrow \text{Zeros}(W, H);$ **for** $i \leftarrow 1$ **to** $N$ **do** // 1. Primal Pass: Standard Binning
     **foreach** *bin index* $(u, v)$ *in support of* $k$ *around* $(x'_i, y'_i)$ **do**
         $d_x \leftarrow (x'_i - u\Delta)/\Delta; \quad d_y \leftarrow (y'_i - v\Delta)/\Delta$      // Norm. distances
         $H[u, v] \leftarrow H[u, v] + w'_i \cdot k(d_x) \cdot k(d_y);$
     **end**
**end**

**if** *Tangents* $\dot{\mathbf{x}}'$ *are provided* **then** // 2. Forward Mode: Tangent Propagation
     $\dot{H} \leftarrow \text{Zeros}(W, H);$ **for** $i \leftarrow 1$ **to** $N$ **do**
         **foreach** *bin index* $(u, v)$ *in support of* $\kappa$ *around* $(x'_i, y'_i)$ **do**
             $d_x \leftarrow (x'_i - u\Delta)/\Delta; \quad d_y \leftarrow (y'_i - v\Delta)/\Delta;$ $val_x \leftarrow \frac{1}{\Delta}\kappa'(d_x) \cdot \kappa(d_y) \cdot \dot{x}'_i;$
             $val_y \leftarrow \kappa(d_x) \cdot \frac{1}{\Delta}\kappa'(d_y) \cdot \dot{y}'_i;$ $\dot{H}[u, v] \leftarrow \dot{H}[u, v] + w'_i \cdot (val_x + val_y);$
         **end**
     **end**
**end**

**if** *Adjoints* $\bar{H}$ *are provided* **then** // 3. Backward Mode: Adjoint Propagation
     $\nabla_{\mathbf{x}'}\mathcal{L} \leftarrow \text{Zeros}(N, 2);$ **for** $i \leftarrow 1$ **to** $N$ **do**
         $g_x \leftarrow 0; \quad g_y \leftarrow 0;$ **foreach** *bin index* $(u, v)$ *in support of* $\kappa$ *around* $(x'_i, y'_i)$ **do**
             $d_x \leftarrow (x'_i - u\Delta)/\Delta; \quad d_y \leftarrow (y'_i - v\Delta)/\Delta;$
             $g_x \leftarrow g_x + \bar{H}[u, v] \cdot w'_i \cdot \frac{1}{\Delta}\kappa'(d_x) \cdot \kappa(d_y);$
             $g_y \leftarrow g_y + \bar{H}[u, v] \cdot w'_i \cdot \kappa(d_x) \cdot \frac{1}{\Delta}\kappa'(d_y);$
         **end**
         $(\nabla_{\mathbf{x}'}\mathcal{L})_i \leftarrow (g_x, g_y);$
     **end**
**end**
**return** $H, \dot{H}, \nabla_{\mathbf{x}'}\mathcal{L}$

**Algorithm 1:** 2D Functional Binning: Primal, Forward Mode (JVP), and Backward Mode (VJP)

---

## 3.4 BIAS ANALYSIS

Before applying the method to complex tasks, we explicitly validate its ability to approximate long-range finite differences. Following the methodology of (Gallego et al., 2019), we proceed with the following steps for an event packet $\{e_k = (x_k, y_k, t_k, p_k)\}_{k=1}^{N_e}$.

1. Events are warped to the mean time using rotational models:
$$T_{rot} : e_k \mapsto (t_{ref} - t)\boldsymbol{\omega} \times \boldsymbol{x}_k + \boldsymbol{x}_k, \qquad \boldsymbol{x}_k = (x_k, y_k, 1) \mapsto \boldsymbol{x}'_k. \tag{15}$$

2. An Image of Warped Events (IWE) of resolution $200 \times 150$ is constructed using non-differentiable (*rect*), differentiable (*linear*), and biased (*gauss*) binning kernels:
$$k_{rect}(x) = \mathbb{1}_{|x|<\frac{1}{2}}, \quad k_{linear}(x) = (1 - |x|)\mathbb{1}_{|x|<1}, \quad k_{gauss}(x) = \frac{1}{\sqrt{2\pi}}e^{-x^2/2}\mathbb{1}_{|x|<\frac{3}{2}}. \tag{16}$$

3. Synthesized gradients are computed with $l(x) = \max(1 - |x|, 0)$ with respect to Variance (Var) (Gallego et al., 2018) and Log-Likelihood (LL) (Gu et al., 2021) scores:
$$\text{Var} = \frac{1}{N_p}\sum_{i,j}(h_{i,j} - \mu_H)^2, \quad \text{LL} = \sum_{i,j}\log(\text{NB}(h_{i,j}|r, p)), \tag{17}$$

where $\mu_H$ is the frame mean, and $(r, p) = (0.3, 0.8)$. Theoretical results show that this rectangular kernel has second-order accuracy in approximating the gradient of a score.

We analyze bias using 20,000 events from the *dynamic_rotation* sequence of the Event Camera Dataset (ECD) (Mueggler et al., 2017). A uniform grid of candidate angular velocities is sampled within $[-5, 5]^3$, yielding 1,331 score evaluations and 3,993 analytic gradients. Finite-difference bias is estimated by subtracting numerical gradients computed via central difference with a step size of 1.0. The results are shown in Figure 3.

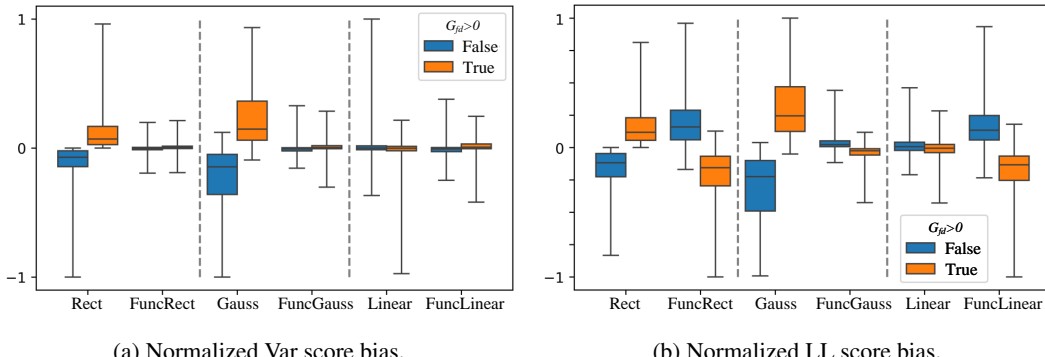

(a) Normalized Var score bias.  (b) Normalized LL score bias.

Figure 3: Bias analysis results. The analytical gradients are subtracted by numerical gradients to obtain the finite-difference bias, **colored by the sign of finite-difference gradients** $G_{fd}$.

As shown in Figure 3(a), the variance score exhibits strong bias with the *rect* and *gauss* kernels due to discontinuities, while the *linear* kernel remains unbiased. Our gradient estimation substantially reduces bias across all kernels, even improving the *linear* kernel by better approximating long-range finite differences.

For the log-likelihood score (Figure 3(b)), results are mixed: *rect* kernel bias is reduced at the cost of higher variance, while the *linear* kernel bias increases. Consistent with our theoretical analysis, this reflects the limited second-order accuracy of the triangular reconstruction $l(x)$ in approximating the nonlinear LL score. This suggests the exact gradient estimation method should be application-dependent, which our framework supports via adaptable cotangent reconstruction constraints.

## 4 EXPERIMENTS

This section further validates the proposed method on event-based motion estimation tasks, specifically angular and linear velocity estimation, where the computed gradients guide the optimization. Besides the rotational motion model defined in Equation (15), we introduce a linear motion model:

$$T_{trans} : e_k \mapsto (t_{ref} - t)\boldsymbol{v} + \boldsymbol{x}_k, \ \boldsymbol{x}_k = (x_k, y_k, 1) \mapsto \boldsymbol{x}'_k. \tag{18}$$

We evaluate all model–kernel–score combinations on eight sequences from the Event Camera Dataset (ECD), namely *boxes, dynamic, poster*, and *shapes* (both rotation and translation). The sequences are partitioned into non-overlapping packets of $N_e = 20,000$ events to estimate motion parameters. Accuracy is quantified using Root Mean Square (RMS) error in °/s for rotation and m/s for translation. Images of Warped Events (IWEs) are constructed from projected coordinates with a resolution of $200 \times 150$ and a binning step of $\Delta = 0.01$. Experiments were conducted on a laptop equipped with an NVIDIA RTX 4090 GPU using the JAX framework (Bradbury et al., 2018). We employed distinct optimizers tailored to the kernel properties: L-BFGS-B (Liu & Nocedal, 1989) for the *rect* and *linear* kernels, and trust-ncg (Steihaug, 1983) for the *gauss* kernel.

### 4.1 OPTIMIZATION RESULTS

The mean performance across all sequences is presented in Figure 4. It is evident that our method improves both accuracy and convergence speed for angular velocity estimation, achieving a 10.3% reduction in RMS error and a $1.66\times$ acceleration in convergence. In the case of linear velocity,

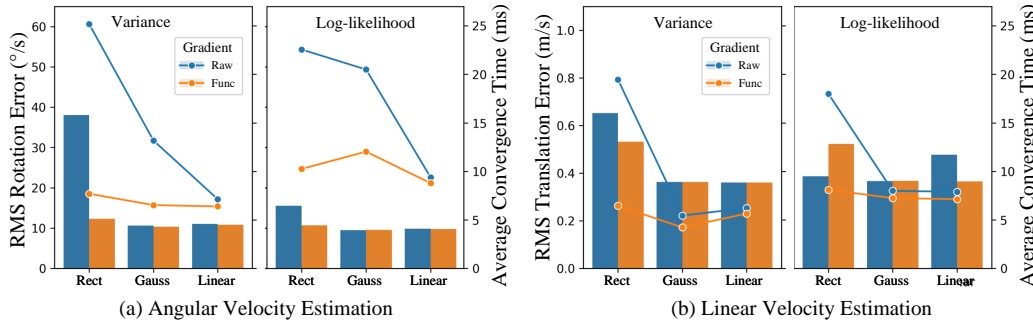

Figure 4: Optimization results for motion estimation, with combined bar charts showing RMS estimation accuracy and line charts showing the mean convergence time for every $N_e = 20000$ events.

singularities of this task (Guo & Gallego, 2024) lead to a marginal 3.9% increase in RMS error, yet the method maintains a $1.48\times$ faster convergence rate. Overall, the proposed method yields a 3.2% improvement in RMS error alongside a $1.57\times$ speedup. Since our approach modifies only the gradients without altering the solution space, these performance gains indicate that our synthesized gradients provide more effective update directions. Although the gradient computation incurs higher theoretical complexity (Equation (13)), the empirical results demonstrate that this cost is outweighed by the benefits in optimization efficiency. For complete results, please refer to the appendix.

## 4.2 ABLATION STUDIES AND BASELINE COMPARISONS

This subsection investigates the efficacy of Functional Backpropagation (FBP) using the linear reconstruction kernel. To isolate the effects of the reconstruction method, we focus on the rotational case utilizing the $k_{rect}$ binning kernel and the Var and LL score function.

**Sensitivity to Reconstruction Kernels.** For the reconstruction kernel $l(x)$, we additionally evaluate the Cubic kernel (assuming a $C^2$ smooth cotangent function) and the Lanczos kernel (assuming a band-limited cotangent function), defined as:

$$l_{Cubic}(x) = \begin{cases} 1.5|x|^3 - 2.5|x|^2 + 1 & |x| < 1 \\ 2.5|x|^2 - 0.5|x|^3 - 4|x| + 2 & 1 < |x| \leq 2 \\ 0 & \text{otherwise} \end{cases} , \quad (19)$$

$$l_{Lanczos}(x) = \frac{2\sin(\pi x)\sin(\frac{\pi}{2}x)}{\pi^2 x^2}\mathbb{1}_{|x| \leq 2}.$$

Table 1: **Ablation study on reconstruction kernels.** Different cotangent reconstruction kernels are compared in angular velocity estimation in Accuracy/Time (°/s and ms). Bold: best value.

| Kernel | boxes_rotation | | dynamic_rotation | | poster_rotation | | shapes_rotation | |
|---|---|---|---|---|---|---|---|---|
| | Var | LL | Var | LL | Var | LL | Var | LL |
| Bicubic | 12.47/7.37 | 10.17/9.17 | 6.70/8.56 | 5.88/10.78 | 14.39/7.61 | 12.48/9.54 | 16.47/11.31 | **13.82**/ 18.07 |
| Lanzcos | 12.46/8.84 | **10.10**/10.12 | 6.72/10.36 | **5.86**/12.37 | **14.15**/9.01 | 12.45/10.66 | 17.93/14.83 | 14.25/21.12 |
| Linear | **12.44/6.50** | 10.14/**8.07** | **6.67/7.44** | 5.88/**9.57** | 14.26/**6.55** | **12.42/8.17** | **15.89/10.26** | 14.26/**15.18** |

As shown in Table 1, the Linear kernel achieves the optimal trade-off between accuracy and efficiency. While higher-order kernels offer greater smoothness, they increase computational runtime with diminishing returns in estimation accuracy. Consequently, we utilize the Linear reconstruction kernel for the remainder of our experiments.

**Comparison vs. Heuristic Surrogate Gradients (SG).** We compare FBP against standard Surrogate Gradients commonly used in Spiking Neural Networks: the Straight-Through-Estimator (STE) (Yin et al., 2019) and the Sigmoid Surrogate (Neftci et al., 2019), defined as:

$$\kappa'_{STE} = -\text{sgn}(x)\mathbb{1}_{|x|<1}, \kappa'_{sigmoid} = \sigma'(10(x + \frac{1}{2}) - \sigma'(10(x - \frac{1}{2})), \sigma(x) = \frac{1}{1 + \exp(-x)}. \quad (20)$$

Table 2: **Comparisons with heuristic surrogate gradients.** Different gradient surrogates are compared in angular velocity estimation in Accuracy/Time (°/s and ms). Bold: Best value.

| Surrogate | boxes_rotation | | dynamic_rotation | | poster_rotation | | shapes_rotation | |
|---|---|---|---|---|---|---|---|---|
| | Var | LL | Var | LL | Var | LL | Var | LL |
| STE | 36.29/7.18 | 14.32/8.88 | 10.92/8.45 | 9.00/10.36 | 29.00/7.38 | 16.22/8.91 | 96.89/ 13.07 | 55.83/ 17.35 |
| Sigmoid | 13.94/7.55 | 10.17/8.49 | 7.12/8.74 | 5.88/9.84 | 15.60/7.70 | 12.51/8.68 | 18.76/11.41 | **14.23**/15.44 |
| FBP (Ours) | **12.44/6.50** | **10.14/8.07** | **6.67/7.44** | **5.88/9.57** | **14.26/6.55** | **12.42/8.17** | **15.89/10.26** | 14.26/**15.18** |

From Table 2, FBP achieves a significantly lower error floor and reduced convergence time compared to heuristic SGs. This confirms that deriving the gradient via integration by parts captures the underlying binning geometry more effectively than imposing arbitrary smooth shapes.

### 4.3 COMPUTATIONAL COMPLEXITY

To supplement the optimization analysis in Section 4.1, we verify the computational cost of the proposed method by comparing the execution time of a single call to the original binning function versus the gradient computation function. As Jacobian-vector products (JVPs) inherit the structure of the forward pass without requiring intermediate storage, we present the computation time for a single JVP call in float32 on both CPU and GPU in Table 3, where our method for different binning kernels is prefixed with Func. The data indicates that the computation time approximately doubles compared to the standard deduced JVP. However, when combined with the optimization results in Section 4.1, the overall efficiency and effectiveness of the method are confirmed.

Table 3: Computation time in microseconds. Our modified gradient computation is shown in **bold.**

| Platforms | CPU (R9-7945HX) | | | GPU (4090-laptop) | | |
|---|---|---|---|---|---|---|
| $N_e$ | 20000 | 50000 | 100000 | 20000 | 50000 | 100000 |
| Rect | 37.1 | 69.9 | 110 | 31.1 | 29.6 | 29.7 |
| **FuncRect JVP** | **635** | **893** | **1440** | **42.3** | **62.4** | **78.8** |
| Gauss | 536 | 820 | 1380 | 44.9 | 61.7 | 75.6 |
| Gauss JVP | 638 | 946 | 1600 | 40.6 | 62.3 | 77.7 |
| **FuncGauss JVP** | **1090** | **2130** | **5890** | **83.5** | **114** | **158** |
| **Overhead** | **1.71×** | **2.25×** | **3.68×** | **2.06×** | **1.83×** | **2.03×** |
| Linear | 473 | 551 | 774 | 26.8 | 37.1 | 55.6 |
| Linear JVP | 440 | 543 | 772 | 28.1 | 37.3 | 51.5 |
| **FuncLinear JVP** | **736** | **1190** | **2120** | **66.8** | **83.1** | **117** |
| **Overhead** | **1.67×** | **2.19×** | **2.75×** | **2.38×** | **2.23×** | **2.27×** |

## 5 APPLICATIONS

To test the applicability of the proposed gradient computation method on modern learning pipelines, we conduct further experiments on the latest event-based optical flow estimation network MotionPriorCMax (Hamann et al., 2024) and SLAM algorithm CMax-SLAM (Guo & Gallego, 2024).

### 5.1 MOTIONPRIORCMAX

MotionPriorCMax (MPC) is the SOTA self-supervised optical flow estimation network that uses the linear binning kernel $k_2$ to construct a contrast loss for learning, which is trained against the latest DSEC (Gehrig et al., 2021a) dataset. We changed the binning kernel to $k_{rect}$ using the modified gradient computation method and retrained the model on eight RTX A6000 GPUs. Reported metrics are End Point Error (EPE), Angular Error (AE), and percentage of outliers with EPE>3px (3PE).

**Results.** We present the average DSEC benchmark performance in Figure 5 where our method improves the EPE error by 9.4%, confirming that the new gradient computation method helps to find the optimal parameters even for a complex neural network. A visualization presented in the right

| Method | EPE ↓ | AE ↓ | 3PE ↓ |
|--------|-------|------|-------|
| MPC | 2.81 | 8.96 | 14.5 |
| Ours | **2.54** | **8.33** | **13.3** |

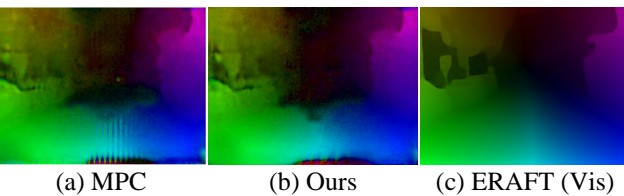

(a) MPC      (b) Ours      (c) ERAFT (Vis)

Figure 5: **Left:** all sequence average performance on DSEC. **Right:** Predicted optical flow (ERAFT only for qualitative visualization). Our method exhibits more robustness with fewer artifacts.

| Method | shapes | | poster | | boxes | | dynamic | |
|--------|--------|-----|--------|-----|-------|-----|---------|-----|
| | Abs | Rel | Abs | Rel | Abs | Rel | Abs | Rel |
| CMax-SLAM | 4.95 | **6.71** | 5.65 | 6.36 | 5.42 | 6.75 | 3.38 | **3.59** |
| Ours | **4.84** | 6.76 | **5.27** | **6.29** | **5.23** | **6.60** | **3.11** | 3.61 |

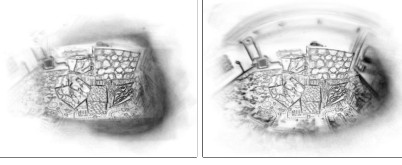

(a) CMax-SLAM      (b) Ours

Figure 6: **Left:** CMax-SLAM results on real-world sequences from ECD. We report Absolute (Abs) and Relative (Rel) trajectory errors (best results in **bold**). **Right:** Mapping result on the full sequence, showing that CMax-SLAM fails on long sequences while our method preserves correctness.

of Figure 5, where optical flow output by ERAFT (Gehrig et al., 2021b) is regarded as ground truth since it is not available on test sequences. It's observed that the network learns more robust features with a sharper IWE, which is only trainable with our method. For a complete result on all 7 test sequences, please refer to the appendix.

## 5.2 CMAX-SLAM

CMax-SLAM (Guo & Gallego, 2024) is the SOTA rotation-only SLAM algorithm that uses the Contrast Maximization framework (Gallego et al., 2018) on both the frontend and backend, where a panoramic IWE is constructed as the map. We changed the binning kernel from $k_{linear}$ to $k_{rect}$ and report the RMS absolute trajectory error (Abs) in [°] and the RMS relative error (Rel) in [°/s] for the first 30s of data of the sequences in ECD (Mueggler et al., 2017) for a fair comparison.

**Results.** The overall ECD benchmark results are shown in Figure 6, where our method mainly improves the Abs RMS error by 5.1% with comparable Rel RMS error. On the right side of Figure 6, we show the reconstruction results of the two methods: the baseline fails on long sequences due to accumulated errors, while our method preserves correctness and yields more robust reconstructions, showing that sharp IWE helps reduce the mapping ambiguity.

## 6 CONCLUSION

In this work, we addressed the long-standing challenge of biased gradient estimation in event binning for event-based vision. By lifting binning functions to the functional space and leveraging weak derivatives through integration by parts, we derived an exact formula for synthesizing unbiased gradients. This framework, termed **Functional Backpropagation**, reconstructs cotangent functions during backpropagation and guarantees unbiasedness without altering the forward outputs. Our theoretical analysis established the correctness and unbiasedness of the synthesized gradients, while extensive experiments on motion estimation, optical flow, and SLAM demonstrated consistent improvements in both accuracy and convergence speed. Looking forward, our framework opens up opportunities to extend unbiased gradient estimation to a wider class of discontinuous operators in neuromorphic computing and to generalizations to nonuniform binning grids and nonlinear reconstruction methods.

ACKNOWLEDGEMENTS

This work is supported by the National Natural Science Foundation of China (NSFC) under Grants 62225207, 62436008, 62306295, and 62576328. The AI-driven experiments, simulations and model training were performed on the robotic AI-Scientist platform of Chinese Academy of Sciences.

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
