## A  APPENDIX

### A.1  USE OF LLMS

This paper uses ChatGPT only for text polishing, correcting mathematical symbols, and searching relevant work. The surrogate gradient (SG) method is one that was new knowledge to the authors and provides useful insights into the presentation of this paper.

### A.2  FUNCTIONAL AUTOMATIC DIFFERENTIATION

This section supplements the discussion in Section 3.1 to cover general automatic differentiation, and to make the text self-contained with the necessary mathematical results proved.

To simplify the expression of functional automatic differentiation, we introduce the following conventions:

- **Vertical bar notation.** For an operator $f$, we write

$$f(u|y) := [f(u)](y),$$

meaning that $f$ takes a function $u \in \mathcal{H}(X)$ as input, produces a function in $\mathcal{H}(Y)$, and then we evaluate this function at $y \in Y$. This notation is non-standard, but convenient for distinguishing between function mapping and function evaluation. Given no risk of ambiguity, we may write $f(y) := f(u|y)$ for brevity.

- **Variational symbol $\delta$.** We use $\delta$ in analogy to Leibniz's differential $d$, but acting on functions instead of scalar values. The quotient

$$\frac{\delta f(y)}{\delta g(x)}$$

denotes the *operator derivative* of $f(y)$ with respect to $g(x)$.

### A.2.1  THE CHAIN RULE OF FUNCTIONALS

Fundamental to automatic differentiation is the decomposition of differentials provided by the chain rule of partial derivatives of composite functions. However, since functionals are functions that consume an entire function to give a value, the input dimension is infinite, so the derivatives cannot be expressed by finite-dimensional Jacobian matrices. To extend the chain rule to functionals, remember that derivatives are unique local linear approximations to the function. This leads to the definition of the Fréchet derivative:

**Definition 2** (Fréchet). *Let $V$ and $W$ be normed vector spaces, and $U \subseteq V$ be an open subset of $V$. A function $f : U \to W$ is called Fréchet differentiable at $x \in U$ if there exists a bounded linear operator $A : V \to W$ such that*

$$\lim_{\|h\|_V \to 0} \frac{\|f(x+h) - f(x) - Ah\|_W}{\|h\|_V} = 0. \tag{21}$$

*If there exists such an operator $A$, it is unique, so we write $Df[x] = A$ and call it the Fréchet derivative of $f$ at $x$. A function $f$ is Fréchet differentiable if $\forall x \in U, Df[x]$ exists.*

For the Fréchet derivative, the well-known Fermat's theorem also applies to identify critical points of a given functional by solving an equation involving the derivative. This result makes it possible to treat the Fréchet derivative like an ordinary derivative when solving an optimization problem.

**Theorem 2** (Fermat). *Let $V$ be a normed vector space and $f : V \to \mathbb{R}$ is a functional that is Fréchet differentiable on $V$. Then if $x_0$ is a point where $f$ has a local extremum, $Df[x_0] = \mathbf{0}$ (the zero functional).*

*Proof.* Without loss of generality, let $x_0$ be the local maxima. Then there exists $\delta > 0$ such that when $\|h\|_V < \delta$,

$$f(x_0 + h) \leq f(x_0). \tag{22}$$

According to Definition 2, we have

$$f(x_0 + h) - f(x_0) - Df[x_0](h) = o(\|h\|_V). \tag{23}$$

For all $v \in V$, we can find a $t$ that satisfy $\|tv\|_V < \delta$, substitute $h$ for $tv$:

$$f(x_0 + tv) - f(x_0) = t \cdot Df[x_0](v) + o(|t|) \le 0. \tag{24}$$

Taking $t \to 0^+$, gives $Df[x_0](v) \le 0$, likewise with $t \to 0^-$, we have $Df[x_0](v) \ge 0$. Thus $Df[x_0](v) = 0, \forall v \in V$ and hence $Df[x_0] = 0$. $\qquad\square$

The chain rule has an analogous form as ordinary derivative, but expressed as composition of linear operators.

**Theorem 3** (The Chain Rule). *Let $U$, $V$ and $W$ be normed vector spaces and $U \xrightarrow{f} V \xrightarrow{g} W$. If $f$ is Fréchet differentiable at $u \in U$ and $g$ is Fréchet differentiable at $f(u) \in V$, then the composite function $h = g \circ f$ is Fréchet differentiable at $u$ with the chain rule*

$$Dh[u] = Dg[f(u)] \circ Df[u]. \tag{25}$$

*Proof.* From Definition 2, we have for a small perturbation $\delta u$ such that $h$ is differentiable at $u + \delta u$,

$$h(u + \delta u) = g(f(u + \delta u)), \tag{26}$$
$$= g(f(u) + Df[u](\delta u) + r_f(\delta u)), \tag{27}$$
$$= h(u) + Dg[f(u)](Df[u](\delta u)) + Dg[f(u)](r_f(\delta u)) + r_g(Df[u](\delta u) + r_f(\delta u)), \tag{28}$$

such that $\lim_{\varepsilon \to 0} \frac{\|r_f(\varepsilon)\|_V}{\|\varepsilon\|_U} = 0$, and $\lim_{\eta \to 0} \frac{\|r_g(\eta)\|_W}{\|\eta\|_V} = 0$. Since $Df[u]$ and $Dg[f(u)]$ are bounded, there exist constants $C_1$ and $C_2$ such that $\|Df[u](\varepsilon)\|_V \le C_1\|\varepsilon\|_U$ and $\|Dg[f(u)](\eta)\|_W \le C_2\|\eta\|_V$. So for the remainder, we have

$$\frac{\|Dg[f(u)](r_f(\delta u))\|_W}{\|\delta u\|_U} \le C_2 \frac{\|r_f(\delta u)\|_V}{\|\delta u\|_U} \to 0 \tag{29}$$

and

$$\frac{\|r_g(Df[u](\delta u) + r_f(\delta u))\|_W}{\|\delta u\|_U} = \frac{\|r_g(Df[u](\delta u) + r_f(\delta u))\|_W}{\|Df[u](\delta u) + r_f(\delta u)\|_V} \frac{\|Df[u](\delta u) + r_f(\delta u)\|_V}{\|\delta u\|_U}, \tag{30}$$
$$\le \frac{\|r_g(Df[u](\delta u) + r_f(\delta u))\|_W}{\|Df[u](\delta u) + r_f(\delta u)\|_V} (C_1 + \frac{\|r_f(\delta u)\|_V}{\|\delta u\|_U}) \to 0. \tag{31}$$

So $h(u + \delta u) = h(u) + (Dg[f(u)] \circ Df[u])(\delta u) + o(\|\delta u\|_U)$. $\qquad\square$

In automatic differentiation, closed-form expressions are required rather than abstract operators. To derive this, we add the additional constraint that all vector spaces are Hilbert spaces, so that we can apply the **Riesz Representation Theorem**. To simplify the expression, we introduce a new notation analogous to Leibniz's notation for operators.

**Theorem 4** (Riesz). *Denote $H(X) = L^2(X) \cap C(X)$ as the space of continuous and square-integrable functions on $X$. Let $H(X) \xrightarrow{f} H(Y) \xrightarrow{g} H(Z)$, $f$ is Fréchet differentiable in $H(X)$ and $g$ is Fréchet differentiable in $H(Y)$. Denote point evaluation functionals on $H(Y)$ and $H(Z)$ as $L_y : v \mapsto v(y), \forall v \in H(Y)$ and $L_z : w \mapsto w(z), \forall w \in H(Z)$ respectively and the Riesz representation of $D(L_y \circ f)[u]$ as $\frac{\delta f(u|y)}{\delta I(u|x)}$, where $I$ represents the identity mapping, then the Riesz representation of composite functional $D(L_z \circ g \circ f)[u]$ is:*

$$\frac{\delta g(f(u)|z)}{\delta I(u|x)} = \int_Y \frac{\delta g(f(u)|z)}{\delta f(u|y)} \frac{\delta f(u|y)}{\delta I(u|x)} dy. \tag{32}$$

*Proof.* From the definition, $L_*$ is a linear operator so $DL_*[\cdot] = L_*$. $D(L_z \circ g \circ f)[u] = L_z \circ Dg[f(u)] \circ Df[u]$ from the chain rule, then $\forall v \in H(X)$, we have

$$L_z \circ Dg[f(u)] \circ Df[u](v) = \int_Y \frac{\delta g(f(u)|z)}{\delta f(u|y)}(Df[u](v))(y)dy, \tag{33}$$

$$= \int_Y \frac{\delta g(f(u)|z)}{\delta f(u|y)}(L_y \circ Df[u])(v)dy, \tag{34}$$

$$= \int_Y \frac{\delta g(f(u)|z)}{\delta f(u|y)}dy \int_X \frac{\delta f(u|y)}{\delta I(u|x)}v(x)dx, \tag{35}$$

$$= \int_{X \times Y} \frac{\delta g(f(u)|z)}{\delta f(u|y)}\frac{\delta f(u|y)}{\delta I(u|x)}v(x)dxdy. \tag{36}$$

Since $v \in H(X)$ is arbitrary, the theorem is proved. $\qquad\square$

### A.2.2 INTERPRETATION OF AUTOMATIC DIFFERENTIATION OF FUNCTIONALS

In ordinary AD, the task is to compute Jacobian-Vector products (forward mode) or Vector-Jacobian products (reverse mode) so that entries of the full Jacobian matrix can be computed by selecting the test vectors to be the standard basis $e^{(i)}$ in the tangent or cotangent spaces. In functional AD, the analogue is the integration of the product of the functional derivative representation and the tangent (cotangent) function.

**Definition 3** (Functional Forward Mode AD). *Given a Fréchet differentiable function $f : H(X) \to H(Y)$ and tangent function at $u \in H(x)$ as $v \in T_u H(X)$, the functional forward mode AD program computes the tangent function at $f(u) \in H(Y)$ as $w \in T_{f(u)}H(Y)$.*

$$w(y) = \int_X \frac{\delta f(u|y)}{\delta I(u|x)}v(x)dx. \tag{37}$$

**Definition 4** (Functional Reverse Mode AD). *Given a Fréchet differentiable function $f : H(X) \to H(Y)$ and cotangent function at $f(u) \in H(Y)$ as $w^* \in T^*_{f(u)}H(Y)$, the functional reverse mode AD program computes the cotangent function at $u \in H(X)$ as $v^* \in T^*_u H(X)$.*

$$v^*(x) = \int_Y \frac{\delta f(u|y)}{\delta I(u|x)}w^*(y)dy. \tag{38}$$

Now, with the formal definition of functional AD, it's possible to derive a recursive procedure to compute the Riesz representation of composite functionals:

**Corollary 1.** *Let $f_i : H(X_i) \to H(X_{i+1})$ be Fréchet differentiable in $H(X_i)$ and $F_n = f_{n-1} \circ f_{n-2} \circ ... \circ f_1$. The functional forward mode AD states that the tangent function of $F_n$ at $F_n(u)$ can be recursively computed as:*

$$w(x_{m+1}) = \int_{X_m} \frac{\delta f_m(F_m(u)|x_{m+1})}{\delta f_{m-1}(F_{m-1}(u)|x_m)}w(x_m)dx_m \tag{39}$$

*and the functional reverse mode AD states that the cotangent function of $F_n$ at $u$ can be recursively computed as:*

$$v^*(x_{m-1}) = \int_{X_m} \frac{\delta f_{m-1}(F_{m-1}(u)|x_m)}{\delta f_{m-2}(F_{m-2}(u)|x_{m-1})}v^*(x_m)dx_m \tag{40}$$

*where we define $f_{-1} = f_0 = I$ as the identity map so $F_0 = F_1 = I$.*

### A.2.3 THE CONNECTION BETWEEN FUNCTIONS AND SAMPLED DIRAC COMBS

In Equation (11), we make an equation between the cotangent function and the sampled cotangent vector. This is understood in the distributional sense. In fact, we have:

**Theorem 5.** *For any smooth function $g(x) \in \mathcal{H}(\mathbb{R})$, if Equation (11) hold, we have:*

$$\lim_{\Delta \to 0} \int g(x)v_h(x)dx = \sum_{j=1}^{W} \int g(x)v_{h_j}\delta(x - j\Delta)dx. \tag{41}$$

*Proof.* Select simple functions $g(x) = \max(1 - |\frac{x-j\Delta}{\Delta}|, 0)$, the equation is proved by observing that the right-hand side constitutes the Riemann sum. As any smooth function can be approximated by summing such simple functions, the result is proved. $\qquad\square$

### A.3 THE UNBIASEDNESS CONSTRAINT

The problem of ensuring unbiasedness is how to quantify the bias when the true gradients don't exist. To achieve that, we resort to line integrals so that the bias can be measured in the distributional sense. Let $\phi$ be an arbitrary smooth function that accepts a binned event frame $\boldsymbol{h} \in H_d$ to return a scalar $\phi(\boldsymbol{h})$, then by the fundamental theorem of calculus for line integrals, we have for every piecewise-smooth curve $\boldsymbol{h} : [a, b] \to H_d$:

$$\phi(\boldsymbol{h}(b)) - \phi(\boldsymbol{h}(a)) = \int_a^b \nabla\phi(\boldsymbol{h}(t)) \cdot \boldsymbol{h}'(t)dt, \tag{42}$$

where $\boldsymbol{h}'(t)$ should be understood in the weak sense. Assuming we want to find the optimal histogram parameters by inferring the extreme values of some smooth objective, then the derivative $\boldsymbol{h}'(t)$ is the only function that makes the above equation true. In this view, if the above equation is nearly true with $\tilde{\boldsymbol{h}}'(t)$, then we call it an unbiased gradient estimation.

For a simplified equation that can be used to check the unbiasedness, we pick each entry $h_j(t)$ inside the vector dot product and approximate $\nabla_j\phi(\boldsymbol{h}(t))$ with another arbitrary smooth function $\varphi_j(t)$, then the above argument translates to:

$$\int \varphi_j(t)h_j'(t)dt = -\int \varphi_j'(t)h_j(t)dt, \tag{43}$$

when $h_j\phi_n$ vanishes at the boundaries. Since there's no $\tilde{\boldsymbol{h}}'(t)$ other than $\boldsymbol{h}'(t)$ to ensure the equality for arbitrary $\varphi_i$ and curve $\boldsymbol{h}$, we only require the equality to hold for a family of simple functions on simple paths. In practice, we select the pow functions and paths on directly related parameters $x'_{i_d}$s. Since both sides of Equation (45) are linear functionals of $\varphi_i$, equality only needs to hold for the elementary power functions. So we have the following definition.

**Definition 5.** *If for all polynomial functions $\varphi_n(t)$ of degree $n$, equality*

$$\int_T \varphi_n(t)\tilde{h}'(t)dt = -\int_T \varphi_n'(t)h(t)dt \tag{44}$$

*holds for all $n \le m$ but not for $n > m$, we say $\tilde{h}'(t)$ estimates the weak derivative $h'(t)$ with a degree of precision of $m$ on path $T$. If $m \ge 1$, we say the estimation is unbiased.*

The definition above quantifies how well the estimation of a weak derivative is. The definition of biasedness conforms to the fact that the derivative is the best local linear approximation. For a multidimensional gradient field approximated by the cumulative product of 1D functions, we have the same result if each function satisfies Definition 5.

It's now easy to derive the unbiasedness constraint of the proposed method. In fact, we have the following result.

**Corollary 2.** *The synthetic weak derivative $\frac{\partial\tilde{h}_j}{\partial x_i^j}$ in Equation (14) is an unbiased estimation of the weak derivative $\frac{\partial h_j}{\partial x_i^j}$ on path $T$ if the reconstruction kernel $l(\cdot)$ satisfies*

$$\int_T k(x)dx = \int_T \int_{\mathbb{R}} l(y)k(x - y)dydx. \tag{45}$$

Corollary 2 provides a minimum requirement for the gradient estimation to be unbiased. However, since the binning function is evaluated on a regular grid, the neighborhood information of $h$ can provide useful information when selecting $l(\cdot)$ instead of a random guess. In many cases, $\phi(\cdot)$ attributes to $h_i$ a similar contribution as $h_j$ if $i$ is in a neighborhood of $j$, then we have $v_{h_i} \approx v_{h_j}$. In other words, we can assume the underlying cotangent function space to be smooth, such that local information provides enough characterization. One simple assumption is that the underlying space should have a minimum slope, then we drive the linear spline kernel $l(x) = \max(1 - |x|, 0)$, which

has a minimum support such that Equation (45) holds for a short path $T$. In fact, It can be proved that the resulting $(l * k)'(x)$ provides a second-order accurate estimate of the weak derivative $k'(x)$ along any path containing $\text{supp}(k) \cap [-1, 1]$, provided that $k(\cdot)$ is symmetric with respect to some vertical axis. We conduct the following experiments on this linear reconstruction kernel.

## A.4 PyTorch/JAX Implementation of FBP

This part provides the detailed implementation in PyTorch and JAX for implementing the proposed FBP.

PyTorch (Paszke et al., 2019) is based on reverse-mode automatic differentiation (VJP). A function with custom AD rules should be implemented as a subclass of torch.autograd.Function as follows.

Listing 1: PyTorch implementation of FBP as custom VJP rules.

```python
import torch

class FunctionalBinning(torch.autograd.Function):
    @staticmethod
    def forward(ctx, xarray, yarray, weights, grid_size):
        ctx.save_for_backward(xarray, yarray, weights)
        # Standard binning with k (Forward pass is unchanged!)
        return k_kernel_binning(xarray, yarray, weights, grid_size)

    @staticmethod
    def backward(ctx, grad_frame):
        xarray, yarray, weights = ctx.saved_tensors
        # Use synthesized transposed derivative with kappa (Eq. 14)
        xarray_dot = kappa_kernel_dx_transposed(\
                grad_frame, xarray, yarray, grid_size)
        yarray_dot = kappa_kernel_dy_transposed(\
                grad_frame, xarray, yarray, grid_size)
        return xarray_dot * weights, yarray_dot * weights, None, None
```

The only extra work to do is to derive the transposed derivative kernel of $\kappa(\cdot)$. JAX (Bradbury et al., 2018) is based on forward-mode automatic differentiation (JVP). A function with custom AD rules should be implemented by decorating it with custom_jvp and implementing the custom JVP rules as follows.

Listing 2: JAX implementation of FBP as custom JVP rules.

```python
import jax

@jax.custom_jvp
def FunctionalBinning(xarray, yarray, weights, grid_size):
    # Standard binning with k (Forward pass is unchanged!)
    return k_kernel_binning(xarray, yarray, weights, grid_size)

@FunctionalBinning.defjvp
def FunctionalBinning_JVP(primals, tangents):
    xarray, yarray, weights, grid_size = primals
    xarray_dot, yarray_dot, *_ = tangents

    # 1. Run standard forward pass
    frame = FunctionalBinning(xarray, yarray, weights, grid_size)

    # 2. Compute synthesized derivative using kappa (Eq. 14)
    # instead of evaluating Dirac deltas.
    frame_dot = kappa_kernel_dx_binning(\
            xarray, yarray, xarray_dot * weights, grid_size) +\
                kappa_kernel_dy_binning(\
```

```
                xarray , yarray , yarray_dot * weights , grid_size )

    return frame , frame_dot
```

## A.5 EXPERIMENTAL DETAILS

We first provide the exact formula of $\kappa_{rect}$, $\kappa_{linear}$, and $\kappa_{gauss}$, which are used to surrogate the gradient rule of the respective kernels $k_{rect}$, $k_{linear}$, and $k_{guass}$:

$$
\kappa_{rect}(x) = \begin{cases} \frac{3}{4} - x^2, & |x| < \frac{1}{2} \\ \frac{1}{8}(3 - 2|x|)^2, & \frac{1}{2} \le |x| < \frac{3}{2} \\ 0. & |x| \ge \frac{3}{2} \end{cases} \tag{46}
$$

$$
\kappa_{linear}(x) = \begin{cases} \frac{1}{6}(4 + 3(-2 + |x|)|x|^2), & |x| < 1 \\ \frac{1}{6}(2 - |x|)^3, & 1 \le |x| < 2 \\ 0. & |x| \ge 2 \end{cases} \tag{47}
$$

$$
\kappa_{gauss}(x) = \begin{cases} \frac{1}{2}(x - 1)\text{erf}\left(\frac{x-1}{\sqrt{2}}\right) - x\text{erf}\left(\frac{x}{\sqrt{2}}\right) \\ \quad + \frac{1}{2}(x + 1)\text{erf}\left(\frac{x+1}{\sqrt{2}}\right) + \frac{e^{-\frac{1}{2}(x+1)^2}\left(e^{2x} - 2e^{x+\frac{1}{2}} + 1\right)}{\sqrt{2\pi}}, & |x| < \frac{1}{2} \\ \frac{1}{2}\left((|x| - 1)\text{erf}\left(\frac{|x|-1}{\sqrt{2}}\right) - 2x\text{erf}\left(\frac{x}{\sqrt{2}}\right) + \text{erf}\left(\frac{3}{2\sqrt{2}}\right)|x| \\ \quad + \text{erf}\left(\frac{3}{2\sqrt{2}}\right) - 2\sqrt{\frac{2}{\pi}}e^{-\frac{x^2}{2}} + \sqrt{\frac{2}{\pi}}e^{-\frac{1}{2}(|x|-1)^2} + \frac{\sqrt{\frac{2}{\pi}}}{e^{9/8}}\right), & \frac{1}{2} \le |x| < \frac{3}{2} \\ \frac{1}{2}(|x| - 1)\text{erf}\left(\frac{x-1}{\sqrt{2}}\right) - \frac{1}{2}\text{erf}\left(\frac{3}{2\sqrt{2}}\right)(|x| - 1) \\ \quad + \frac{e^{-\frac{1}{2}(|x|-1)^2}}{\sqrt{2\pi}} - \frac{1}{e^{9/8}\sqrt{2\pi}}, & \frac{3}{2} \le |x| < \frac{5}{2} \\ 0. & |x| \ge \frac{5}{2} \end{cases} \tag{48}
$$

Although the expressions are definite, they are scaled to match the exact binning function, so they are adaptive to the specific scene.

## A.6 ADDITIONAL EXPERIMENTAL RESULTS

Due to space limitations, detailed results for the analysis and applications are presented here. Table 4 and Table 5 provide the optimization results for all the sequences, where our methods are in bold with optimizer L-BFGS-B suffixed by 1 and trust-ncg suffixed by 2. These tables demonstrate another benefit of our method, which is to provide second-order information to facilitate optimization.

For additional optical flow estimation results, refer to Table 6 and Figure 7. The results are consistent with the main text that our method improves the overall robustness.

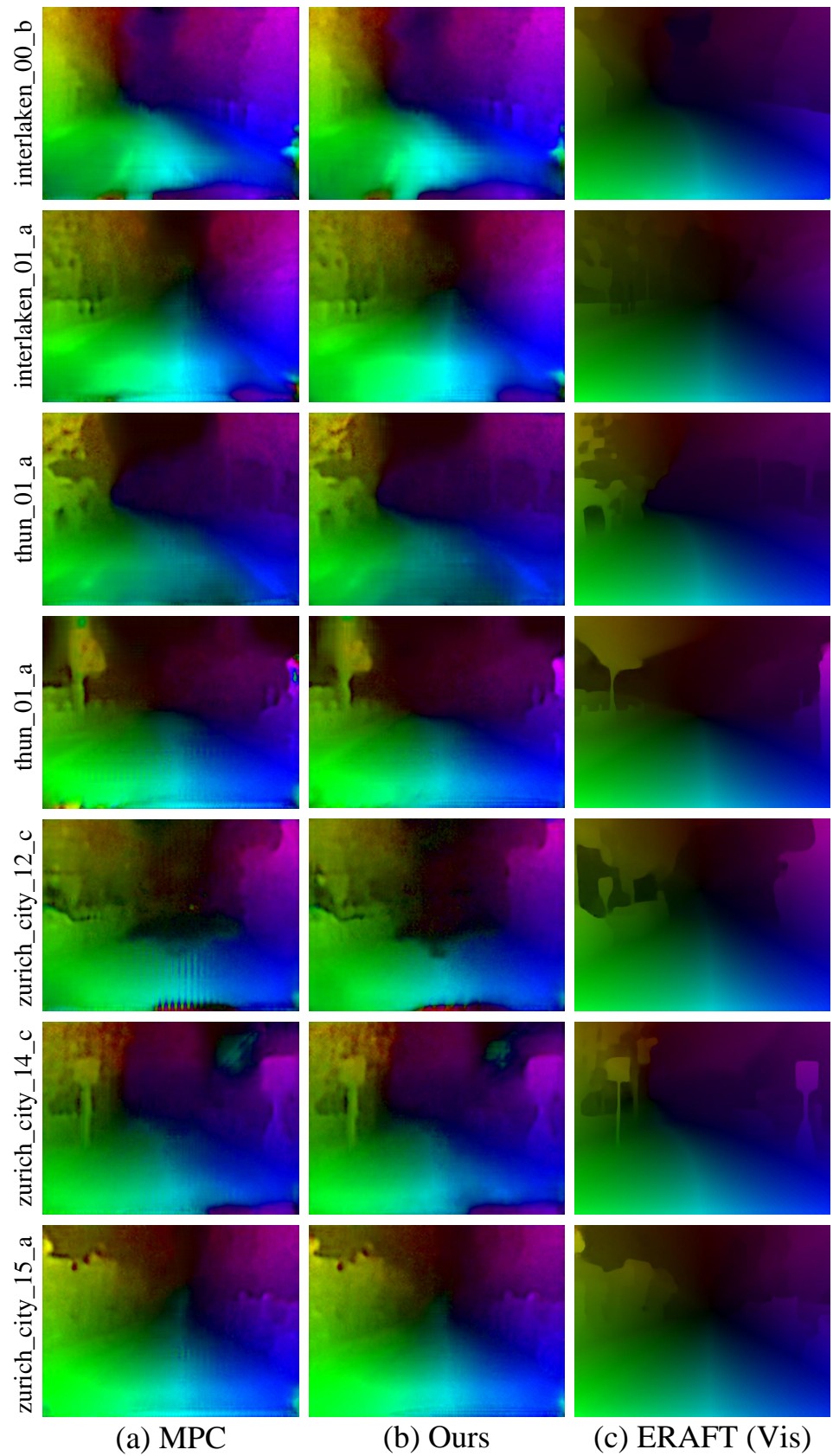

Figure 7: Optical flow estimation results on all 7 test sequences of DSEC (Gehrig et al., 2021a).

Table 4: Angular velocity estimation results. Accuracy in °/s. The optimization time is in milliseconds. n/a means the optimization process has early stops due to abnormal function values.

| Score | Methods | boxes | | dynamic | | poster | | shapes | |
|---|---|---|---|---|---|---|---|---|---|
| | | acc | time | acc | time | acc | time | acc | time |
| Var | Rect | 44.61 | 22.42 | 16.03 | 23.38 | 47.09 | 22.60 | 44.47 | 32.34 |
| | **FuncRect 1** | 12.44 | 6.50 | 6.67 | 7.44 | 14.26 | 6.55 | 15.89 | 10.26 |
| | **FuncRect 2** | 10.91 | 3.00 | 6.11 | 4.24 | 12.83 | 3.30 | 23.94 | 9.96 |
| | Linear | 10.21 | 6.38 | 5.63 | 6.60 | 12.58 | 6.36 | 15.82 | 9.18 |
| | **FuncLinear 1** | 9.82 | 5.26 | 5.53 | 5.82 | 12.16 | 5.39 | 15.75 | 9.06 |
| | **FuncLinear 2** | 9.66 | 2.61 | 5.50 | 4.77 | 12.01 | 3.19 | 14.93 | 13.07 |
| | Gauss 1 | 10.93 | 7.14 | 6.59 | 8.38 | 12.51 | 7.41 | 14.29 | 11.13 |
| | Gauss 2 | 9.61 | 5.01 | 6.22 | 8.36 | 10.80 | 5.49 | 15.79 | 33.87 |
| | **FuncGauss 1** | 10.27 | 6.91 | 6.22 | 7.98 | 11.91 | 7.22 | 13.81 | 11.45 |
| | **FuncGauss 2** | 9.95 | 2.46 | 5.98 | 5.59 | 11.59 | 3.26 | 13.85 | 14.78 |
| LL | Rect | 17.47 | 21.02 | 8.58 | 21.50 | 17.45 | 20.00 | 18.62 | 27.73 |
| | **FuncRect 1** | 10.14 | 8.07 | 5.88 | 9.57 | 12.42 | 8.17 | 14.26 | 15.18 |
| | **FuncRect 2** | 10.95 | 4.31 | 8.48 | 8.73 | 13.80 | 5.67 | n/a | n/a |
| | Linear | 9.04 | 7.29 | 5.33 | 8.93 | 11.53 | 7.98 | 13.36 | 13.30 |
| | **FuncLinear 1** | 8.77 | 6.74 | 5.21 | 7.97 | 11.30 | 6.71 | 13.74 | 13.72 |
| | **FuncLinear 2** | 8.70 | 3.29 | 5.21 | 7.35 | 11.22 | 4.05 | 14.62 | 34.21 |
| | Gauss 1 | 8.99 | 8.68 | 5.64 | 10.28 | 10.97 | 8.80 | 13.46 | 15.89 |
| | Gauss 2 | 7.93 | 5.95 | 5.36 | 12.10 | 9.97 | 6.85 | 14.54 | 57.10 |
| | **FuncGauss 1** | 8.89 | 8.64 | 5.49 | 10.17 | 10.91 | 8.59 | 13.37 | 16.75 |
| | **FuncGauss 2** | 8.64 | 3.80 | 5.35 | 9.51 | 10.74 | 5.30 | 13.41 | 29.52 |

Table 5: Linear velocity estimation results. Accuracy in m/s. The optimization time is in milliseconds. n/a means the optimization process has early stops due to abnormal function values.

| Score | Methods | boxes | | dynamic | | poster | | shapes | |
|---|---|---|---|---|---|---|---|---|---|
| | | acc | time | acc | time | acc | time | acc | time |
| Var | Rect | 0.924 | n/a | 0.447 | n/a | 0.326 | 19.45 | 0.913 | n/a |
| | **FuncRect 1** | 0.875 | n/a | 0.209 | 7.93 | 0.304 | 6.45 | 0.739 | n/a |
| | **FuncRect 2** | 0.679 | 3.40 | 0.208 | 5.18 | 0.304 | 3.55 | 0.257 | 10.73 |
| | Linear | 0.680 | 6.28 | 0.208 | 6.75 | 0.304 | 6.22 | 0.250 | 8.33 |
| | **FuncLinear 1** | 0.683 | 5.63 | 0.208 | 6.78 | 0.303 | 5.64 | 0.249 | 8.16 |
| | **FuncLinear 2** | 0.683 | 3.48 | 0.208 | 6.77 | 0.303 | 3.71 | 0.250 | 12.73 |
| | Gauss 1 | 1.069 | n/a | 0.209 | 8.14 | 0.803 | n/a | 0.796 | n/a |
| | Gauss 2 | 0.689 | 5.26 | 0.209 | 8.67 | 0.305 | 5.44 | 0.249 | 24.64 |
| | **FuncGauss 1** | 0.688 | 6.84 | 0.209 | 8.81 | 0.304 | 7.02 | 0.248 | 11.45 |
| | **FuncGauss 2** | 0.689 | 3.85 | 0.209 | 8.38 | 0.305 | 4.22 | 0.249 | 15.35 |
| LL | Rect | 0.763 | 18.34 | 0.217 | 20.05 | 0.308 | 18.00 | 0.258 | 25.84 |
| | **FuncRect 1** | 0.681 | 7.97 | 0.432 | n/a | 0.303 | 8.10 | 0.673 | n/a |
| | **FuncRect 2** | 0.683 | 6.00 | 0.218 | 12.12 | 0.304 | 6.28 | 0.663 | n/a |
| | Linear | 0.684 | 7.91 | 0.215 | 10.11 | 0.304 | 7.89 | 0.706 | n/a |
| | **FuncLinear 1** | 0.685 | 6.96 | 0.215 | 9.61 | 0.304 | 7.13 | 0.257 | 14.01 |
| | **FuncLinear 2** | 0.685 | 6.00 | 0.216 | 11.47 | 0.304 | 5.30 | 0.256 | 28.93 |
| | Gauss 1 | 1.108 | n/a | 0.450 | n/a | 0.304 | 8.76 | 0.861 | n/a |
| | Gauss 2 | 0.689 | 7.65 | 0.217 | 15.62 | 0.305 | 7.99 | 0.257 | 50.15 |
| | **FuncGauss 1** | 0.688 | 8.59 | 0.218 | 12.34 | 0.710 | n/a | 0.795 | n/a |
| | **FuncGauss 2** | 0.689 | 6.72 | 0.218 | 15.39 | 0.305 | 7.24 | 0.257 | 31.05 |

Table 6: Comparison of MPC and MPC (w/ FBP) across multiple sequences. Metrics are End-Point Error (EPE), Angular Error (AE), and percentage of outliers (%Out).

| Method | All | | | interlaken_00_b | | | interlaken_01_a | | | thun_01_a | | |
|---|---|---|---|---|---|---|---|---|---|---|---|---|
| | EPE | AE | %Out | EPE | AE | %Out | EPE | AE | %Out | EPE | AE | %Out |
| MPC | 2.81 | 8.96 | 14.48 | 3.33 | 5.11 | 19.41 | 2.42 | 6.17 | 16.15 | 1.48 | 6.70 | 8.33 |
| MPC (w/ FBP) | 2.54 | 8.33 | 13.31 | 3.26 | 5.05 | 18.96 | 2.36 | 5.70 | 15.70 | 1.42 | 6.39 | 7.52 |

| Method | thun_01_b | | | zurich_city_12_a | | | zurich_city_14_c | | | zurich_city_15_a | | |
|---|---|---|---|---|---|---|---|---|---|---|---|---|
| | EPE | AE | %Out | EPE | AE | %Out | EPE | AE | %Out | EPE | AE | %Out |
| MPC | 1.47 | 5.88 | 8.18 | 5.98 | 21.56 | 21.02 | 1.97 | 9.74 | 15.25 | 1.54 | 6.71 | 9.25 |
| MPC (w/ FBP) | 1.45 | 5.64 | 7.67 | 4.71 | 19.65 | 17.67 | 1.90 | 9.22 | 13.82 | 1.50 | 6.22 | 7.95 |