# OpenReview forum: "Unbiased Gradient Estimation for Event Binning via Functional Backpropagation"
_ICLR.cc/2026/Conference — ICLR 2026 Poster_

### Official Review · Reviewer_nBtx · 2025-10-18

**Soundness:** 3
**Presentation:** 3
**Contribution:** 3
**Rating:** 6
**Confidence:** 4

**Summary:**

This paper proposed a framework for unbiased gradient estimation of arbitrary binning functions by synthesizing weak derivatives during backpropagation while keeping the forward output unchanged. Experiments show that the proposed method is benificial for downstream tasks like optical flow estimation and SLAM.

**Strengths:**

- The proposed method addresses event-based visual perception from a new perspective via biased gradient estimation.
- The proposed method improves simple optimization-based egomotion estimation and facilitates fast convergence.
- The proposed method is beneficial for downstream tasks like optical flow and SLAM.

**Weaknesses:**

- If it is possible, please evaluate your proposed method on more flow estimation methods, like event-based meshflow estimation.
- The proposed method has many design choices and parameter setups. If it is possible, please enrich the parameter analyses and conduct more detailed ablation studies to help better understand the design choices. Besides, it would be nice to evaluate the proposed method under different event data representations.

**Questions:**

- Would you consider making the source code and the implementation publicly available to foster future research in this line?
- The event-based optical flow estimation has the advantage of temporal-continuous prediction. How about your solution for temporal-continuous prediction? This could be discussed in detail.

---

> ### Author Response · Authors · 2025-11-26
>
> **Meshflow Evaluation:**
> We appreciate the suggestion, as event-based meshflow is indeed a promising direction. However, a primary challenge is that existing meshflow algorithms rely exclusively on frame-level features, whereas our FBP framework operates at the event level. Consequently, incorporating event-level features would require a fundamental redesign of the underlying modeling approach, which falls outside the scope of this paper. We have included a discussion on this limitation and identified it as a key area for future work.
>
> **Different Event Representations:**
> The reviewer requested evaluation under different representations. We clarify that the choice of the binning kernel ($k$) *is* the representation:
>
>   * **Rect Kernel ($k_{rect}$):** Corresponds to standard Voxel Grids or Event Count images.
>   * **Linear Kernel ($k_{linear}$):** Corresponds to bilinearly interpolated representations.
>     By validating our method across these kernels (Section 4), we effectively cover the most common grid-based representations used in modern pipelines.
>
> **Source Code & Continuous Prediction:**
>
>   * **Code:** We commit to releasing the full JAX source code to foster future research.
>   * **Continuous Prediction:** Our method is natively suitable for temporal-continuous prediction because it operates on exact timestamps $t_i$ rather than fixed frames. The functional derivative is defined over the continuous domain, allowing direct optimization of continuous-time trajectories (e.g., splines).

---

> > ### Comment · Reviewer_nBtx · 2025-11-26
> > **Comment**
> >
> > The reviewer would like to thank the authors for their response and clarifications. We would like to maintain the positive rating.

---

### Official Review · Reviewer_vVog · 2025-10-31

**Soundness:** 3
**Presentation:** 1
**Contribution:** 2
**Rating:** 2
**Confidence:** 4

**Summary:**

This paper aims to address a long-standing core problem in event-based vision: the biased gradient estimation caused by the discontinuity of the binning operation. Traditional methods, such as using smooth kernels or surrogate gradients, introduce bias during backpropagation, leading to inefficient learning. To solve this, the authors propose a novel framework named "Functional Backpropagation" (FBP). The core idea is to lift the discrete binning function into a continuous functional space. By leveraging weak derivatives and integration by parts from functional analysis, the FBP framework derives a synthesized, unbiased gradient estimate during backpropagation without altering the forward pass output. The authors theoretically prove that this synthesized gradient approximates long-range finite differences. Experimental results show that the proposed method achieves improved performance on downstream tasks such as motion estimation, optical flow, and SLAM.

**Strengths:**

The paper is built on a very solid mathematical foundation. The authors skillfully employ tools from functional analysis, such as weak derivatives and integration by parts, to provide a novel and theoretically unbiased solution to the prevalent problem of gradient estimation for discontinuous binning operations in the event-based vision domain. This approach of lifting a discrete problem into a functional space for a solution is highly inspiring and represents a rigorous attempt to fundamentally address the gradient issue.

**Weaknesses:**

1. The experimental figures and setup descriptions are insufficient. The readability of some figures is poor. The legend in Figure 3 (e.g., G_fd > 0) is not explained in the caption. The plots on the left of Figure 4 are missing a y-axis title. The experimental setup in Section 4.1 (Bias Analysis) is not clearly described. The authors mention estimating the "finite-difference bias" by subtracting "numerical gradients" but do not detail the specific method for calculating these numerical gradients (e.g., central difference, forward difference) or the choice of step size.

2. The paper's readability could be improved. The paper introduces numerous mathematical symbols and theories from functional analysis (e.g., Fréchet derivative, Riesz representation theorem), which may be unfamiliar to many researchers in the machine learning and computer vision communities, significantly raising the barrier to entry. Despite the novelty of the method, the paper lacks a standalone algorithm section or pseudocode to guide implementation. It is difficult for readers to discern from the theoretical derivations in Section 3 how "Functional Backpropagation" is actually implemented in modern deep learning frameworks like PyTorch or JAX. This significantly hinders the quick understanding and reproduction of the work.

3. The depth of the experimental validation are insufficient. The choice of the reconstruction kernel $l(\cdot)$ is crucial to the method. The authors select a linear spline kernel in Section 3.3 but do not provide an ablation study to validate the effect of other choices, leaving the superiority of the current selection unsupported by experimental evidence. While the paper compares FBP with gradients from the raw kernel, a more crucial comparison would be against other established gradient estimation techniques in the field, such as various Surrogate Gradients (SG) methods.

4. The current experimental setup fails to sufficiently demonstrate whether FBP offers a significant advantage over these existing techniques.The reported performance improvement is limited. In several applications, the improvement brought by the method is relatively modest (e.g., a 9.4% EPE reduction in optical flow and a 5.1% Abs RMS error reduction in SLAM). The authors primarily apply the method to frameworks based on Contrast Maximization (CM). However, the CM method itself suffers from several other well-known bottlenecks, such as high sensitivity to initialization, slow convergence, and a tendency to converge to degenerate solutions. While the proposed method theoretically improves the gradient, its limited practical impact weakens the motivation for targeting "gradient bias" as the most critical problem to solve within the CM framework.

**Questions:**

1. Formally, Eq. (14) seems to just replace the original binning kernel $k(\cdot)$ with a new kernel $\kappa(\cdot)$ formed by the convolution of $l(\cdot)$ and $k(\cdot)$. Could the authors provide a more intuitive explanation as to why the gradient derived from the FBP framework is theoretically "unbiased", whereas directly using a smooth kernel (like a Gaussian) is "biased"? Where does its fundamental advantage lie?

2. In the experiments for Eq. (20), the authors use truncated kernels. Is the fundamental source of gradient bias the artificial truncation of the kernel, or is it the discrete sampling of the IWE space? If it is the former, would using an untruncated kernel (or increasing its bandwidth) significantly reduce the bias? If it is the latter, does this imply that increasing the IWE resolution (i.e., decreasing $\Delta$) is a more direct solution than optimizing the gradient itself?

3. The derivation of Eq. (12) relies on approximating the Dirac delta function with a reconstruction kernel $l(\cdot)$. The authors chose a linear spline kernel in their experiments. What is the specific impact of the choice of $l(\cdot)$ on the unbiasedness and variance of the final gradient? Does an "optimal" choice for $l(\cdot)$ exist in theory?

4. In Section 4.2, the authors use optimizers like L-BFGS-B and trust-ncg, which rely on first- or even second-order information. In real-time applications like robotics and SLAM, efficient (quasi-)Newton methods are crucial. Could the authors comment on whether the FBP framework naturally supports efficient computation of second-order derivatives (i.e., Hessian-vector products, HVP)? Does it have advantages or face new challenges in this regard compared to traditional surrogate gradient methods?

---

> ### Author Response · Authors · 2025-11-26
>
> **Figures and Experimental Setup:**
> We appreciate the detailed feedback on clarity.
>
>   * **Figure 3 Legend:** The legend $G_{fd} > 0$ refers to the sign of the numerical gradients. As noted in lines 366-367, data points are "colored by the sign of numerical gradients." We have made this explicit in the revised figure.
>   * **Figure 4 Y-axis:** We have increased the font size and clarity of the y-axis titles ("RMS Rotation Error ($^\circ/s$)" and "RMS Translation Error (m/s)").
>   * **Section 4.1 Setup:** We clarified that the grid was sampled within $[-5, 5]^3$ with a step size of 1. The "finite-difference bias" was calculated by subtracting numerical gradients computed via central difference with a step size of 1.0.
>
> **Theoretical Complexity & Implementation:**
> We understand the concern regarding the heavy mathematical terminology. To address this, we have added a **Algorithm/Pseudocode** section (Common Response, Section 1) showing that the method is implemented as a standard custom autograd function. We also point to Appendix A.3, which provides the exact, implementable formulas for the derived kernels ($\kappa_{rect}$, etc.).
>
> **Comparison with Surrogate Gradients:**
> We have added a direct comparison against standard Surrogate Gradients (STE, Sigmoid) in the revised manuscript (Common Response, Section 2B). FBP outperforms these heuristics by providing a mathematically consistent gradient derived from the binning geometry.
>
> **Magnitude of Improvements & CM Framework:**
>
>   * **Significance:** We respectfully argue that a **9.4% reduction in EPE** on the competitive DSEC benchmark and a **5.1% reduction in SLAM trajectory error** are significant improvements for a method that strictly alters the *gradient computation* without changing the model architecture or data resolution.
>   * **Contrast Maximization (CM):** While CM has known bottlenecks (sensitivity to initialization, slow convergence), the fact that our method achieves these gains *within* the CM framework proves that **gradient bias was a critical limiting factor**. By providing better update directions, FBP directly mitigates the "slow convergence" issues you highlighted (1.57$\times$ speedup).
>
> **Unbiasedness vs. Smooth Kernels (Eq. 14):**
> You asked why FBP is theoretically better than simply using a smooth kernel (like a Gaussian) in the forward pass. The distinction is fundamental:
>
> 1.  **Forward Pass Fidelity:** Standard smooth binning (Gaussian) **blurs the input data**, permanently losing high-frequency temporal information and altering the physical meaning of the event stream. FBP maintains the **sharp, discontinuous binning** (Rect/Linear) in the forward pass, preserving exact data.
> 2.  **Mathematical Consistency:** "Unbiasedness" in our framework is defined via the fundamental theorem of calculus for line integrals. A gradient is unbiased if its line integral matches the finite difference of the objective function. We prove that our derived kernel $\kappa$ satisfies this condition (Section 3.3), whereas heuristic surrogates do not.
>
> **The Effect of the Reconstruction Kernel**
> We have added the experiments concerning the effect of different reconstruction kernels. The results demonstrates that as long as the unbiasedness constraint is satisfied, the accuracy only changes by little, while for heuristic surrogates the difference is huge, so we choose the linear kernel in the experiments mainly for speed. An optimal reconstruction kernel exists as it comes from the inherent smoothness of the cotangent function. However, the exact smoothness constraint is also dependent on the target function, which may lead to complex and nonlinear reconstruction methods. In this paper we choose to satisfies the unbiasedness constraint while keeping the reconstruction kernel simple for general optimization and learning algorithms.
>
> **Source of Gradient Bias (Eq. 20):**
> The bias arises from the **discontinuity** of the binning function, which generates Dirac deltas that cannot be sampled by standard backpropagation. While increasing IWE resolution (decreasing $\Delta$) reduces discretization error, it does not solve the non-differentiability. FBP bridges this gap at finite resolutions by using signal reconstruction to recover the information lost by the discontinuity, avoiding the need for infinite resolution.
>
> **Second-Order Optimization:**
> Yes, the FBP framework naturally supports second-order optimization. Because the derived gradient (Eq. 14) is itself a well-defined function, it allows for the computation of Hessian-vector products (HVP). Our experiments utilized `trust-ncg` (a Newton-CG optimizer), confirming that FBP provides valid curvature information, which contributed to the 1.66$\times$ faster convergence in angular velocity estimation.

---

> > ### Comment · Reviewer_vVog · 2025-11-28
> >
> > I thank the authors for their detailed response and the additional experiments provided during the rebuttal phase. These materials have clarified several points that were unclear in the original submission.
> > I recognize the value of this work, particularly the elegant mathematical derivations (e.g., utilizing kernel properties in Eq. 13 and the integration-by-parts trick in Eq. 12) to overcome the biased gradient estimation inherent in the original Contrast Maximization (CM) framework. In light of this, I have decided to tentatively raise my rating to 4.
> > However, I still have some remaining concerns that I would like to discuss with the authors. My final decision on whether to further increase the score will depend on the outcome of this discussion.
> >
> > 1. As mentioned in my initial review, the CM method itself suffers from well-known bottlenecks, such as sensitivity to initialization and slow convergence. While this paper aims to address the gradient bias within CM, it appears more like a "patch" for the CM framework rather than a fundamental solution. A truly principled approach to resolving these issues would likely involve designing a discrepancy measure directly on the warped events, thereby avoiding the binning process into image frames entirely. The "unbiased gradient" proposed here is unbiased only relative to the finite difference of the Image of Warped Events (IWE) loss. Since the IWE loss itself relies on binning (discretization), even an "unbiased" gradient is not derived from the intrinsic continuous event stream geometry. Consequently, the motivation feels somewhat constrained, as the method's theoretical ceiling is limited by the IWE representation itself.
> >
> > 2. There appear to be inconsistencies between Figure 3 and Figure 4 that currently prevent me from being fully convinced that the proposed method is a better choice over classic approaches.
> > While Figure 3(a) demonstrates that the method is unbiased, Figure 3(b) shows that under certain kernels, the performance actually degrades. This is counter-intuitive.
> > Figure 4 seems to suggest that as long as a differentiable kernel (like Linear or Gaussian) is used, the optimization converges well, even if the gradient is theoretically biased. The Linear kernel, in particular, appears to offer an excellent balance of efficiency and accuracy. This paradoxically suggests that a simple Linear kernel might be sufficient for most practical applications. If the complex Functional Backpropagation (FBP) framework does not demonstrate a decisive advantage over these simpler baselines, its practical necessity becomes questionable.
> >
> > 3.Some Minor Issues
> > - Legends are missing in Fig. 3(b) and Fig. 4(b). Fig. 4(a) is missing the label for the right y-axis.
> > - The layout for FuncGauss JVP and FuncLinear JVP in Table 3 is confusing; they appear as separate methods but the row alignment is ambiguous. Furthermore, the speedup associated with JVP (Jacobian-Vector Product) makes me confused. The paper states that the proposed method is theoretically slower than classical methods, but the speedup presented here give the impression that it is actually faster.
> > - The notation regarding dimensions is inconsistent throughout the paper. Eq. 5 is multi-dimensional, Eqs. 6-14 are derived in 1D, and Section 3.3 discusses the 2D case. Unifying these would improve clarity.
> > - The core derivation skips several important steps. For instance, does Eq. 12 rely on integration-by-parts? I strongly suggest reorganizing the theoretical section into a formal "Theorem-Proof" structure to make the logic more accessible and rigorous.

---

> > > ### Author Response · Authors · 2025-12-02
> > >
> > > We thank the reviewer for their detailed engagement with our work and for recognizing the elegance of the mathematical derivations involved in overcoming biased gradient estimation. We appreciate the tentative decision to raise the rating to 4. Below, we address your remaining concerns regarding the scope of the Contrast Maximization (CM) framework, specific experimental inconsistencies, and the theoretical derivations.
> > >
> > > ## 1. Motivation: CM Framework vs. Event-by-Event Metrics
> > > You raised a valid point that CM suffers from initialization sensitivity and that our method might appear as a "patch" rather than a fundamental solution like defining discrepancy directly on warped events.
> > > - **Necessity of Binning:** While event-by-event processing is theoretically appealing, the vast majority of downstream computer vision tasks (e.g., optical flow, recognition) rely on dense frame representations to leverage conventional image processing pipelines (e.g., CNNs). Our method bridges the gap between raw asynchronous events and these powerful grid-based tools, and demonstrates that in our application experiments.
> > > - **Fundamental Theoretical Value:** Far from being just a "patch" for CM, this work addresses a "broader issue in learning with discontinuous nonlinearities in neuromorphic computing". Spiking neuron models inherently introduce non-differentiable functions; our approach provides a principled way to compute unbiased gradients for these discontinuities via integration by parts, rather than relying on heuristic surrogate gradients.
> > >
> > > ## 2. Inconsistencies in Figure 3 (Bias Analysis)
> > > You noted that Figure 3(b) shows performance degradation for the Log-Likelihood (LL) score under certain kernels, which seems counter-intuitive compared to Figure 3(a).
> > > - **Predicted by Theory:** This behavior is actually consistent with our theoretical analysis (see previous Section 3.3 and revised Appendix A.3). The "unbiased" property relies on the reconstruction kernel matching the test function. As noted in lines 352-354, the triangular reconstruction kernel $l(x)$ used has "limited second-order accuracy... in approximating the nonlinear LL score".
> > > - **Application Dependence:** Figure 3(a) (Variance score) represents the "correct" case where the reconstruction works ideally. The mixed results in 3(b) highlight that the exact gradient estimation is application-dependent. However, for the primary task of motion estimation using Variance (the standard for CM), the bias reduction is substantial.
> > >
> > > ## 3. The "Paradox" of the Linear Kernel
> > > You observed that the simple Linear kernel often converges well, questioning the practical necessity of the complex FBP framework.
> > > - **Enabling Sharp Representations:** The Linear kernel forces the use of a smooth, "blurry" event frame. A key advantage of our method is that it allows the optimization of sharp kernels (like `Rect`) which are typically non-differentiable. Sharp IWEs help reduce mapping ambiguity and are crucial for robust reconstruction, as shown in our SLAM results.
> > > - **Second-Order Optimization:** While we used first-order optimizers in the main text for fair comparison, our method facilitates second-order optimization. As shown in the Appendix (Tables 4 and 5), FBP provides valid second-order information that further facilitates optimization, which is impossible for the Linear Kernel that is not second-order differentiable.
> > > - **Convergence vs. Latency:** You mentioned confusion regarding speedups. While the single-step computation time of FBP is theoretically higher (approx. 2x slower per JVP call) , the convergence speed is significantly faster (1.57x speedup overall) because the unbiased gradients provide more effective update directions, requiring fewer iterations.
> > >
> > > ## 4. Clarifications on Minor Issues
> > > - **Legends & Layouts:** We have fixed the missing legends in Fig. 3(b) and Fig. 4(b), and the y-axis label in Fig. 4(a). The layout for Table 3 has been adjusted to clearly align "FuncGauss JVP" and "FuncLinear JVP".
> > > - **Unified Dimensions:** We acknowledge the notation shift. The core derivation is presented in 1D ($D=1$) to simplify the "intuitive derivation", but the binning function definition (Eq. 5) and the implementation (Section 3.3) support arbitrary dimensions. We have added a clarifying note connecting the 1D derivation to the 2D implementation.
> > > - **Derivation Steps & Theorem-Proof Structure:** Yes, Equation 12 relies explicitly on integration by parts. We also agree that a formal structure improves rigor. But due to space limitations in the main text, the formal "Theorem-Proof" structure regarding the chain rule of functionals and unbiasedness is detailed in Appendix A.2 (Theorems 2, 3, 4) and Appendix A.3. We have added pointers in the main text to these proofs.

---

### Official Review · Reviewer_2yXt · 2025-10-31

**Soundness:** 2
**Presentation:** 2
**Contribution:** 3
**Rating:** 6
**Confidence:** 2

**Summary:**

The paper contributes a theoretical framework for unbiased gradient estimation in event-based vision, addressing the problem of discontinuities in event binning functions. The authors derive weak derivatives via integration by parts and demonstrate improvements in several tasks such as motion estimation, optical flow, and SLAM.

**Strengths:**

1. The paper is clearly written and well-organized.
2. The mathematical exposition is detailed and precise, with good use of notation and references.
3. Figures and tables are informative and visually consistent, effectively supporting the text.

**Weaknesses:**

1. No ablation on reconstruction kernel choices.
2. Limited comparison to related works, like comparison to prior gradient approximation methods.
3. Readability could be improved with intuitive explanations or schematic examples.

**Questions:**

1. Although convergence speed improves, the computational overhead of functional derivatives is briefly mentioned but not rigorously profiled. A breakdown of runtime cost per gradient step would be helpful.
2. How sensitive is the method’s performance to the choice of reconstruction kernel? Would different kernels change the unbiasedness or convergence behavior?

---

> ### Author Response · Authors · 2025-11-26
>
> **Reconstruction Kernel Ablation:**
> We have addressed your concern regarding the choice of reconstruction kernel with a new **Ablation Study** (see Common Response, Section 2A). This study verifies that the **Linear kernel** provides the best balance of accuracy and efficiency compared to Cubic and Sinc kernels.
>
> **Computational Overhead:**
> We have integrated the detailed runtime breakdown (Forward vs. Backward vs. Optimization time) from Appendix A.5 into the main text. This rigorously profiles the cost-benefit ratio, showing that faster convergence compensates for the heavier backward pass.
>
> **Readability:**
> We hope the newly added **Pseudocode** (see Common Response, Section 1) provides the "intuitive explanation" you requested, bridging the gap between the functional analysis theory and practical coding.

---

### Official Review · Reviewer_4sPS · 2025-10-31

**Soundness:** 3
**Presentation:** 3
**Contribution:** 3
**Rating:** 4
**Confidence:** 3

**Summary:**

The paper addresses the biased gradient estimation in event-based vision due to binning function discontinuities. The proposed functional backpropagation framework lifts binning functions into functional space to enable unbiased gradient computation via weak derivatives without Dirac deltas. The paper claims that the weak derivative is equivalent to long-range finite differences and generalizes to both smooth and discontinuous kernels. Experiments are conducted on velocity estimation, optical flow estimation and SLAM.

**Strengths:**

1. The topic is good. The paper analyzes a fundamental issue in event cameras, namely the gradient problem in the binning function, which is a relatively small but important issue.
2. The paper features rigorous mathematical derivations and proofs, demonstrating a solid theoretical foundation.
3. The method proposed in this paper has the potential to contribute to some fundamental applications in the field, such as velocity estimation as mentioned in the paper.

**Weaknesses:**

1. Some captions of the figures are insufficient, which affects the readability of the paper. For Figures 1 and 2, it is recommended to add more detailed explanations of the symbols and the content depicted in the figures. This will help readers better understand the key components and the underlying concepts illustrated in these figures.
2. The current structure of the paper writting is not well balanced. I believe the method and analysis sections in the early part are overly lengthy, while the experimental section in the main text is clearly too brief.
3. In the part of velocity estimation, the proposed methods show no advantages in some cases in Linear Velocity Estimation.
4. There are some issues with the optical flow estimation section. First, are the comparison methods used in the paper up-to-date? This needs to be carefully verified. The related work section cites mostly older references. Second, what are the advantages of the proposed method compared to other existing deep learning algorithms? Can deep learning models implicitly address the effects caused by binning within the network computation? Third, I think the visualization of the method's result is not satisfying, and using the ERAFT inference results as ground truth is also not reasonable enough.

**Questions:**

As listed in the "Weaknesses".

---

> ### Author Response · Authors · 2025-11-26
>
> **Figures and Captions:**
> We agree that the captions for Figures 1 and 2 were insufficient. We have rewritten them to explicitly map the symbols ($v_{hd}$, $v_p$) to the functional backpropagation flow described in Section 3.1, clarifying the "Sampling" and "Reconstruction" steps visually.
>
> **Paper Organization:**
> We clarify that the "Analysis" section serves as the primary experiment to demonstrate the theoretical properties (unbiasedness) and convergence speed of FBP on the fundamental task of motion estimation. The "Applications" section validates scalability on complex downstream tasks. We have reorganized the paper to make this clearer.
>
> **Linear Velocity Estimation:**
> You correctly noted the lack of accuracy advantage in Linear Velocity estimation ($3.9\%$ higher error). This stems from inherent singularities in the flow field for this specific motion. However, we emphasize that FBP still yields a **1.48$\times$ speedup** in convergence time. For real-time robotics applications, this trade-off between marginal error increase and significant speed gain is often favorable.
>
> **Optical Flow Baselines and ERAFT:**
> We clarify that **ERAFT was used solely for qualitative visualization** in Figure 5 because the ground truth for the DSEC test set is withheld by the benchmark organizers.
>
>   * **Quantitative Metrics:** The reported **9.4% EPE reduction** is computed against the official DSEC benchmark server, confirming the improvement over the baseline.
>   * **Baseline Choice:** We respectfully clarify that MotionPriorCMax is the current state-of-the-art (SOTA) for unsupervised learning on the DSEC dataset, making it the most rigorous baseline for our method. Our contribution is not a new network architecture, but a plug-and-play gradient estimator that improves existing SOTA pipelines. The older references in Section 2 refer to foundational binning concepts, not the downstream tasks.
>   * **On Deep Learning & Binning:** Deep networks cannot implicitly "solve" the non-differentiability of binning because standard backpropagation requires a defined gradient at the binning layer. Current models rely on smooth approximations, but our method enables sharp representations. Such replacement is beneficial as demonstrated by the 9.4% reduction in End Point Error (EPE) on the DSEC benchmark.

---

### Author Response · Authors · 2025-11-26
**Common Response to All Reviewers**

We thank the reviewers for their constructive feedback and for recognizing the theoretical novelty and potential for faster convergence of our method. We acknowledge the consensus regarding the mathematical density and the need for clearer comparisons. Below, we outline the major revisions, which are colored in red in the paper.

## 1 Pseudocode & Implementation (Addressing R2, R3)
To lower the barrier to entry noted by Reviewers **vVog** and **2yXt**, we have added a standalone algorithm section in the main text and the PyTorch/JAX code in the appendix. This demonstrates that despite the heavy functional analysis theory, the practical implementation is a lightweight custom autograd function compatible with modern frameworks like JAX and PyTorch.

## 2 New Experiments: Ablations & Comparisons (Addressing R2, R3, R4)

To address concerns regarding the choice of reconstruction kernels and the lack of baselines, we have added the following experiments to the revised manuscript.

**A. Ablation Study: Sensitivity to Reconstruction Kernels**
We evaluate three reconstruction kernels $l(x)$ for Equation (12) on the ECD rotational sequences:

1.  **Cubic:** Assumes a $C^2$ smooth cotangent function. $l(x)=\begin{cases}(a+2)|x|^3-(a+3)|x|^2+1 & |x|<1\\\\a|x|^3-5a|x|^2+8a|x|-4& 1<x\leq 2\\\\0&\text{otherwise}\end{cases}$, and $a=-0.5$.
2.  **Lanzcos:** Assumes a band-limited cotangent function. $l(x)=\begin{cases}\frac{a\sin(\pi x)\sin(\pi x/a)}{\pi^2x^2} & |x|<a\\\\0 & \text{otherwise}\end{cases}$, and $a=2$.
3.  **Linear (Ours):** Assumes a piecewise linear cotangent function. $l(x)=\max(1-|x|,0)$

**Results (Accuracy in $^\circ/s$, Time in ms):**

|  Kernel | boxes_rotation | | dynamic_rotation | | poster_rotation | | shapes_rotation | |
|:-------:|----------------|---------------|------------------|--------------|-----------------|---------------|-----------------|---------------|
| | Var | LL | Var | LL | Var | LL | Var | LL |
| Bicubic | 12.47/7.37   | 10.17/9.17  | 6.70/8.56      | 5.88/10.78 | 14.39/7.61    | 12.48/9.54  | 16.47/11.31   | 13.82/18.07 |
| Lanzcos | 12.46/8.84   | 10.10/10.12 | 6.72/10.36     | 5.86/12.37 | 14.15/9.01    | 12.45/10.66 | 17.93/14.83   | 14.25/21.12 |
| Linear  | 12.44/6.50   | 10.14/8.07  | 6.67/7.44      | 5.88/9.57  | 14.26/6.55    | 12.42/8.17  | 15.89/10.26   | 14.26/15.18 |

*Conclusion:* The **Linear kernel** offers the optimal trade-off between accuracy and computational cost. While higher-order kernels (Cubic and Lanzcos) are theoretically smoother, they provide diminishing returns in accuracy while increasing runtime.

**B. Comparison vs. Heuristic Surrogate Gradients (SG)**
Addressing **vVog**, we compare Functional Backpropagation (FBP) against standard SGs common in SNNs. In all cases, the **forward pass remains the sharp Rect binning**.

1.  **Straight-Through-Estimator (STE):** Hard proxy derivative. $k'(x)=\begin{cases}-1 & 0<x<1\\\\1 &-1<x<0 \\\\ 0 & \text{otherwise}\end{cases}$
2.  **Sigmoid Surrogate:** Smooth derivative approximation ($\sigma'$). $k'(x)=\sigma'(k(x+\frac{1}{2}))-\sigma'(k(x-\frac{1}{2}))$, with $\sigma(x)=\frac{1}{1+\exp(-x)}$ and $k=10$.
3.  **FBP (Ours):** Mathematically derived weak derivative.

**Results:**

| Surrogate | boxes_rotation | | dynamic_rotation | | poster_rotation | | shapes_rotation | |
| :--- | :---: | :---: | :---: | :---: | :---: | :---: | :---: | :---: |
| | Var | LL | Var | LL | Var | LL | Var | LL |
| STE | 36.29/7.18 | 14.32/8.88 | 10.92/8.45 | 9.00/10.36 | 29.00/7.38 | 16.22/8.91 | 96.89/13.07 | 55.83/17.35 |
| Sigmoid | 13.94/7.55 | 10.17/8.49 | 7.12/8.74 | 5.88/9.84 | 15.60/7.70 | 12.51/8.68 | 18.76/11.41 | 14.23/15.44 |
| FBP (Ours) | 12.44/6.50 | 10.14/8.07 | 6.67/7.44 | 5.88/9.57 | 14.26/6.55 | 12.42/8.17 | 15.89/10.26 | 14.26/15.18 |

*Conclusion:* FBP achieves a significantly lower error floor than heuristic SGs. This confirms that deriving the gradient via integration by parts matches the specific binning geometry better than imposing an arbitrary smooth shape (like a Sigmoid).

## 3 Computational Cost Analysis

We have moved the runtime profiling from Appendix A.5 to the main text. While the backward pass is approximately $2\times$ slower per step due to the convolution operations, the method converges **1.57$\times$ faster overall** (wall-clock time), justifying the added complexity per iteration.

## 4 Reorganized Presentation & Typos

We have moved the proof of the unbiasedness in Section 3.3 to the appendix and brought forward the bias analysis to our methodology, for relieving the readers from the dense math while concentrating on the main results. The former Analysis section is renamed to Experiments with additional ablation studies, comparisons and computational cost analysis, to prevent the readers from treating it as the extended mathematical derivations. A few reference typos and inaccurate descriptions are also fixed, shown in red color in the main text.

---

### Meta-Review · Area_Chair_NGiQ · 2026-01-06

**Summary:**

The paper addresses the biased gradient estimation in event-based vision due to binning function discontinuities. It proposes "Functional Backpropagation", which lifts binning functions to a functional space, to solve the issue. The paper initially received two 6 ratings and one 4 rating and one 2 rating (vVog). After initial discussion, vVog decides to tentatively upgrade the score to rating 4 with additional questions to be further clarified. The reviewers acknowledged: 1) its theoretical novelty, rigouring mathematical derivations and proofs; 2) has the potential to be applied to other applications in the field. The main raised concerns are 1) its computational overhead (but with faster convergence speed overall); 2) advantages of the method over other existing deep learning algorithms. By considering its novel and rigorous theoretical contribution, improved accuracy of downstream applications, as well as improved overall computational efficiency, the AC is happy to recommend the acceptance of the work.

**Reviewer Concerns:**

The AC think the critical concerns have been addressed.

**Reviewer Scores:**

The AC think reviewer 4sPS and vVog would upgrade their scores if further discussions are conducted.

---

### Decision · Program_Chairs · 2026-01-26

Accept (Poster)